# Clonal kinetics and single-cell transcriptional profiling of CAR-T cells in patients undergoing CD19 CAR-T immunotherapy

Alyssa Sheih[1,8], Valentin Voillet [2,8], Laïla-Aïcha Hanafi[1,8], Hannah A. DeBerg[3], Masanao Yajima[4], Reed Hawkins[1], Vivian Gersuk [3], Stanley R. Riddell[1,5,6], David G. Maloney[1,5,6], Martin E. Wohlfahrt [1], Dnyanada Pande[1], Mark R. Enstrom [1], Hans-Peter Kiem [1,5,7], Jennifer E. Adair [1,5,6], Raphaël Gottardo [2,5,6], Peter S. Linsley[3] & Cameron J. Turtle[1,5,6]*

Chimeric antigen receptor (CAR) T-cell therapy has produced remarkable anti-tumor responses in patients with B-cell malignancies. However, clonal kinetics and transcriptional programs that regulate the fate of CAR-T cells after infusion remain poorly understood. Here we perform TCRB sequencing, integration site analysis, and single-cell RNA sequencing (scRNA-seq) to profile CD8$^+$ CAR-T cells from infusion products (IPs) and blood of patients undergoing CD19 CAR-T immunotherapy. TCRB sequencing shows that clonal diversity of CAR-T cells is highest in the IPs and declines following infusion. We observe clones that display distinct patterns of clonal kinetics, making variable contributions to the CAR-T cell pool after infusion. Although integration site does not appear to be a key driver of clonal kinetics, scRNA-seq demonstrates that clones that expand after infusion mainly originate from infused clusters with higher expression of cytotoxicity and proliferation genes. Thus, we uncover transcriptional programs associated with CAR-T cell behavior after infusion.

[1] Clinical Research Division, Fred Hutchinson Cancer Research Center, Seattle, Washington 98109, USA. [2] Vaccine and Infectious Disease Division and Public Health Sciences Division, Fred Hutchinson Cancer Research Center, Seattle, Washington 98109, USA. [3] Benaroya Research Institute at Virginia Mason, Seattle, Washington 98101, USA. [4] Department of Mathematics and Statistics, Boston University, Boston, Massachusetts 02215, USA. [5] Department of Medicine, University of Washington, Seattle, Washington, USA. [6] Integrated Immunotherapy Research Center, Fred Hutchinson Cancer Research Center, Seattle, Washington 98109, USA. [7] Department of Pathology, University of Washington, Seattle, Washington, USA. [8] These authors contributed equally: Alyssa Sheih, Valentin Voillet, Laïla-Aïcha Hanafi. *email: cturtle@fredhutch.org

Lymphodepletion chemotherapy followed by infusion of CD19-specific chimeric antigen receptor modified-T (CAR-T) cells has produced durable responses in a subset of patients with relapsed and refractory B-cell malignancies[1–8]. In vivo CAR-T cell proliferation after infusion is important for anti-tumor efficacy and may depend on several factors including the design of the CAR construct, the quality of T cells for CAR T manufacturing, the manufacturing process, the lymphodepletion regimen, infused cell phenotype, and the tumor burden and tumor microenvironment[3,4,9–18]. Despite the importance of in vivo CAR-T cell expansion to clinical response and toxicities of CAR-T cell therapy, little is known about the clonal composition of CAR-T cells in the infusion product (IP), how clonal composition of CAR-T cells changes in the recipient after adoptive transfer, and how distinct transcriptional signatures in the CAR-T IP might affect cell fate in vivo.

We previously reported in a small subset of patients with acute lymphoblastic leukemia (ALL) that CAR-T cells were polyclonal, both in the IP and at the peak of expansion in the recipient[4]. Another group reported one patient in whom a single CD8[+] CAR-T cell clone, in which the transgene had integrated into the TET2 locus, dominated at the peak of in vivo expansion[19]. These highly disparate patterns suggest variability in the clonal composition of infused CAR-T cells and potential differences in the ability of individual CAR-T cell clones to expand after adoptive transfer. Thus, we examine the T cell receptor beta (TCRB) repertoire and lentiviral integration sites of CD8[+] CAR-T cells isolated from the IP and from blood of patients treated with CD19-targeted CAR-T cell immunotherapy. We find distinct patterns of clonal behavior that contribute to the CAR-T cell population in the recipient after infusion. Using single-cell RNA sequencing (scRNA-seq), we identify transcriptionally distinct clusters of infused CD8[+] CAR-T cells that differ in their contribution to the CAR-T cell repertoire in blood after infusion.

## Results

**Clonal diversity of CAR-T cells decreases after infusion.** To better understand changes in the composition of CAR-T cells after infusion, we studied a cohort of patients ($n = 10$) who received CD19-specific CAR-T cells manufactured from bulk CD4[+] T cells and CD8[+] central memory-enriched (T$_{CM}$) cells, which were infused in a 1:1 ratio of CD4[+]:CD8[+] CAR-T cells. These patients were representative of the study population in terms of age, sex, lymphodepletion therapy, cell dose, adverse events, and clinical outcome (Supplementary Table 1). Analysis of CAR-T cell expansion following infusion showed that CD8[+] CAR-T cells reached peak counts between 1 and 2 weeks post infusion and declined thereafter (Fig. 1a). To study how the CD8[+] CAR-T cell repertoire changes during in vivo expansion and contraction, we performed high-throughput sequencing of the TCRB CDR3 region in CD8[+] CAR-T cells isolated from the IPs, from blood early after infusion (day 7–14, early), and from blood after the peak of CAR-T cell expansion (day 26–30, late)[20]. We found that multiple TCRB gene families were represented in CD8[+] CAR-T cells isolated from the IP and from blood after infusion (Supplementary Fig. 1). Examination of the unique TCRB sequences confirmed that CD8[+] CAR-T cells were highly polyclonal in the IP and after adoptive transfer (Supplementary Table 2).

We evaluated the similarity of the TCRB repertoire between CD8[+] CAR-T cells in the IP and in the blood at early and late time points using the Morisita index[21]. A Morisita index of 0 indicates no TCRB sequences are shared between two samples. An index of 1 indicates that the repertoires are identical, comprising the same clonotypes in the same proportions in each sample. In all patients, we observed sharing of CAR-T cell clonotypes between the IP and after adoptive transfer (Fig. 1b). TCRB repertoire similarity was greater between CD8[+] CAR-T cells in the IP and those isolated early after infusion compared with those isolated at the later time point (Fig. 1b), consistent with increasing differences in the CD8[+] CAR-T cell repertoire over time after adoptive transfer.

Next, we analyzed changes in the clonal diversity of CD8[+] CAR-T cells after adoptive transfer by calculating the Shannon entropy index after adjusting for differences in sampling depth[22]. Clonal diversity of CD8[+] CAR-T cells was highest in the IP and progressively declined over time in the blood (Fig. 1c). In contrast, clonal diversity remained stable over time in EGFRt[−] endogenous CD8[+] T cells. As time after infusion increased, the ten most prevalent clonotypes comprised an increasing proportion of the CD8[+] CAR-T cell repertoire (Fig. 1d), demonstrating that decreased clonal diversity after infusion is partly due to expansion of oligoclonal CAR-T cell populations.

**CAR-T cells display different clonal kinetics after infusion.** To obtain insight into the contribution of individual CAR-T cell clones in the IP to the in vivo CAR-T cell pool, we first determined whether the ten most prevalent clonotypes in the IP remained dominant after infusion. We tracked the rank and relative frequency of these clonotypes within the CD8[+] CAR-T cell repertoire over time (Fig. 2a, b). In several patients, such as ALL-2 and ALL-3, a majority of the top ten clonotypes in the IP remained among the top ten clonotypes after infusion, maintaining a high relative frequency over time. In other patients, both the rank and relative frequency of many of the top ten clonotypes decreased considerably after infusion, suggesting that not all prevalent clonotypes in the IP will contribute to the CAR-T cell pool after infusion.

Next, we traced the origin of the ten most prevalent clonotypes from the early and late time points (Fig. 3a, b). Several of the top ten clonotypes from the early and late time points originated from low-ranking clonotypes in the IPs whose relative frequency greatly increased after infusion. Interestingly, the top ten clonotypes from the early time point generally remained among the top ten clonotypes at the late time point in most patients, except in NHL-1, NHL-3, and NHL-5. In these patients, several of the top ten clonotypes from the early time point decreased in relative frequency and rank at the late time point (Figs. 2b and 3a). Instead, a different subset of CAR-T cell clonotypes, which were initially of lower rank in the IP and at the early time point, emerged as one of the most prevalent clonotypes at the late time point (Fig. 3b). These findings highlight distinct differences in the ability of infused CAR-T cells to either proliferate, survive, and/or remain in blood after adoptive transfer.

Tracking the rank and relative frequency of CAR-T cell clonotypes over time describes changes in the behavior of CAR-T cell clones relative to each other but does not indicate whether the absolute number of any given CAR-T cell clone is increasing or decreasing in the blood. Therefore, we examined the absolute count of individual CD8[+] CAR-T cell clones in blood and observed three patterns of clonal kinetics through the first 30 days after infusion (Fig. 3c). We observed clones whose absolute counts in blood progressively increased, clones that exhibited an early and transient increase followed by a subsequent decrease, and clones that progressively decreased after infusion. We did not identify correlations between the clinical characteristics and patterns of CAR-T cell clonal kinetics. The data indicates that after infusion of a CAR-T cell product manufactured from bulk CD4[+] T cells and CD8[+] T$_{CM}$ cells, there are CD8[+] CAR-T cell clones with different kinetics of accumulation in blood or loss

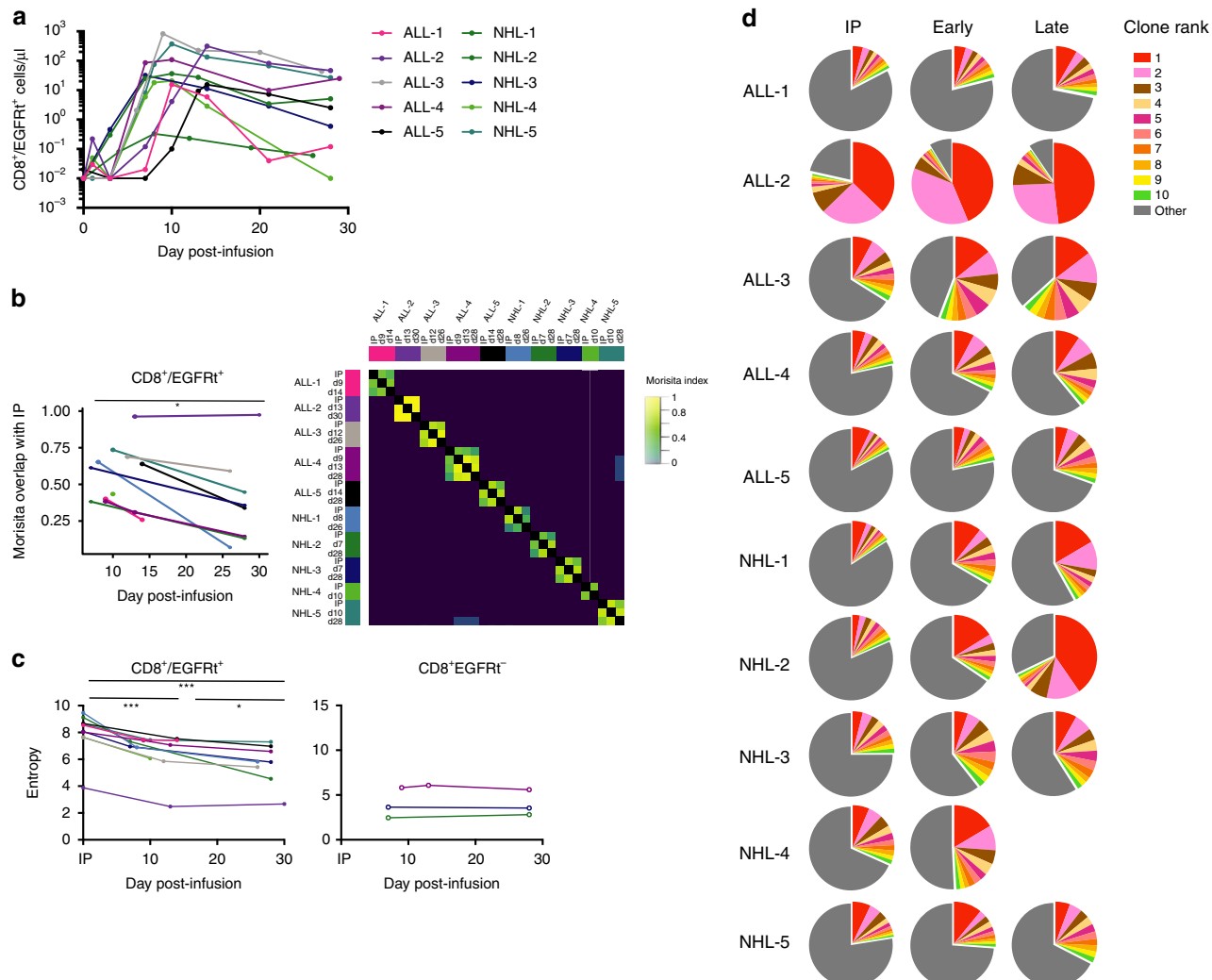

**Fig. 1 Clonal diversity of CD8$^+$ CAR-T cells decreases after infusion. a** CD8$^+$/EGFRt$^+$ CAR-T cell counts in the blood of patients who received CAR-T cells manufactured from CD4$^+$ and CD8$^+$ T$_{CM}$ cells (*n* = 10 patients). Data represents the mean ± SEM. ALL acute lymphoblastic leukemia, NHL non-Hodgkin lymphoma. **b** The Morisita index was calculated for pairwise comparisons of the CD8$^+$ CAR-T cell repertoire between all samples. Left, Morisita overlap index between the IP and the indicated times after adoptive transfer. Each line represents data from an individual patient. Right, heatmap of the Morisita index. The Morisita index was lower at late compared to early times after infusion (paired Mann–Whitney test, *p* < 0.05). **c** Decreased clonal diversity in CD8$^+$ CAR-T cells after infusion. The Shannon entropy indices of the CD8$^+$/EGFRt$^+$ (CAR-T cell, left) and CD8$^+$/EGFRt$^-$ (non-CAR-expressing T cell, right) TCRB repertoires are shown. Statistical differences between samples were evaluated with paired *t*-tests. **p* < 0.05, ***p* < 0.005, ****p* < 0.001. **d** The relative frequencies of the top ten ranked clonotypes in the total CD8$^+$ CAR-T cell TCRB repertoire in the IP and at the early and late time points after infusion are presented in pie charts. The top ten clonotypes are ranked by the number of copies of the TCRB sequence in each sample; the same color might not represent the same identical clonotype at each time point. Source data underlying Fig. 1a, c, and d are provided as a Source Data file.

from the blood after infusion. This could reflect cell intrinsic differences in their capacity to proliferate, survive, or migrate to tissues, or different levels of CAR expression on individual cells that may dictate whether a cell is capable of recognizing tumor targets and the effects of the tumor recognition on cell fate.

**Integration site is not a dominant driver of clonal kinetics.** Vector integration site might be associated with clonal kinetics. Therefore, we analyzed the lentiviral integration profile of CD8$^+$ CAR-T cells isolated from the IPs and from blood after adoptive transfer in ALL (*n* = 3) and non-Hodgkin lymphoma (NHL) (*n* = 4) patients. A total of 55,382 unique integration events (sites) were identified across all patients and samples. The observed integration sites were consistent with lentiviral integration patterns previously described in human T-cell lines[23].

Approximately 82.6% of all sites (45,771) were within genes and integration in introns was more frequent than in exons (Fig. 4a). Consistent with our findings from TCRB sequencing, different clonotypes defined by integration site exhibited distinct in vivo kinetic patterns after CAR-T cell infusion (Fig. 4b). We examined whether these changes in clonotype abundance were associated with distinct genomic loci of vector integration. We identified two different groups of integration sites, sites that either increased or decreased in relative abundance by at least 5-fold between the IP and after infusion. Comparison of genes harboring these integration sites revealed enrichment in many of the same biological pathways (Supplementary Fig. 2). However, genes in biological pathways associated with lymphocyte activation, T cell receptor (TCR) signaling, and regulation of type 1 interferon were more enriched among genes harboring integration sites that increased in relative abundance, and genes associated with T-cell

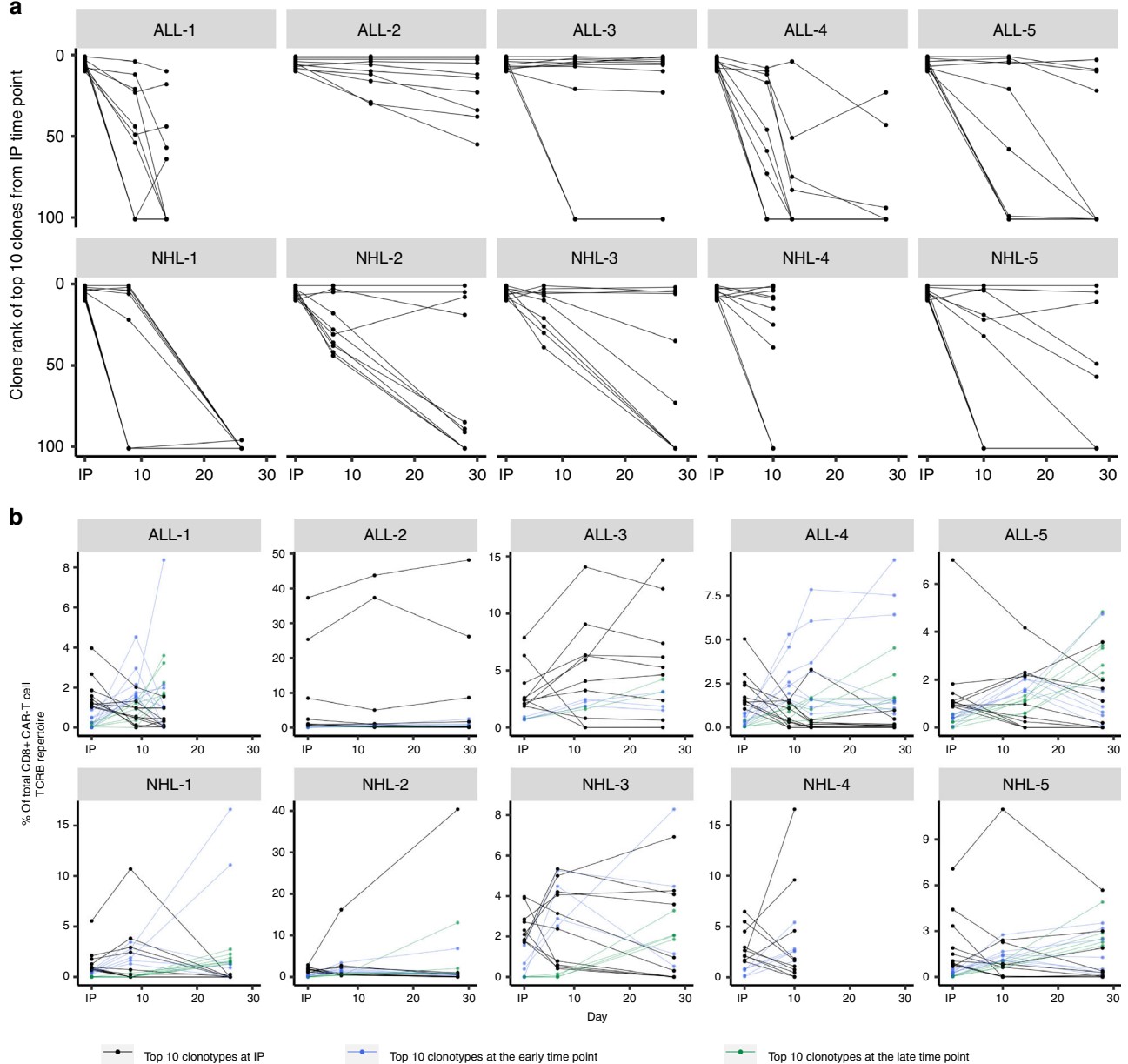

**Fig. 2 Rank and relative frequency of the top ten clonotypes in the IP can change after infusion. a** The rank of the top ten CD8$^+$ CAR-T cell clonotypes in the IP were tracked over time and shown at the early and late time points. All rank positions >100 were designated in the figure as having a rank position of 101. Each line represents an individual clonotype. **b** The relative frequencies of the top ten ranked CD8$^+$ CAR-T cell clonotypes from the IP (black line), the early (blue line), and late (green line) time points were followed over time. Each line represents an individual clonotype. Source data are provided as a Source Data file.

differentiation and the cellular response to UV were more enriched among those harboring integration sites that decreased in relative abundance (Supplementary Fig. 2). These data likely reflect differences in genes that are active at the time of lentiviral transduction.

A recent report identified dominance of a single infused CAR-T cell clone in a single patient associated with integration into the *TET2* gene. Although integrations in the *TET2* gene were observed in our analyses (12 sites in 6 patients), none of these integration sites were among the top 20 most abundant sites identified in any patient or sample, indicating that integration within the *TET2* gene was not a key and frequent driver of clonal expansion in our study. Furthermore, in two patients with highly dominant TCRB clonotypes after infusion (ALL-2 and NHL-2),

we did not identify single integration sites that were responsible for clonal dominance. No integration sites were found at a frequency as high as that of the dominant TCRB clonotype. The most dominant TCRB clonotypes in blood from ALL-2 and NHL-2 at the early time point were 46.0% and 16.8%, respectively. In contrast, in the same samples the highest frequency integration sites in each patient only represented 2.75% and 5.2% of the total integration sites, respectively. These data suggest that an integration site is unlikely to be the key driver of clonal kinetics in our study.

**Single-cell transcriptome analysis of CAR-T cells over time.** The different kinetic behaviors displayed by individual CD8$^+$

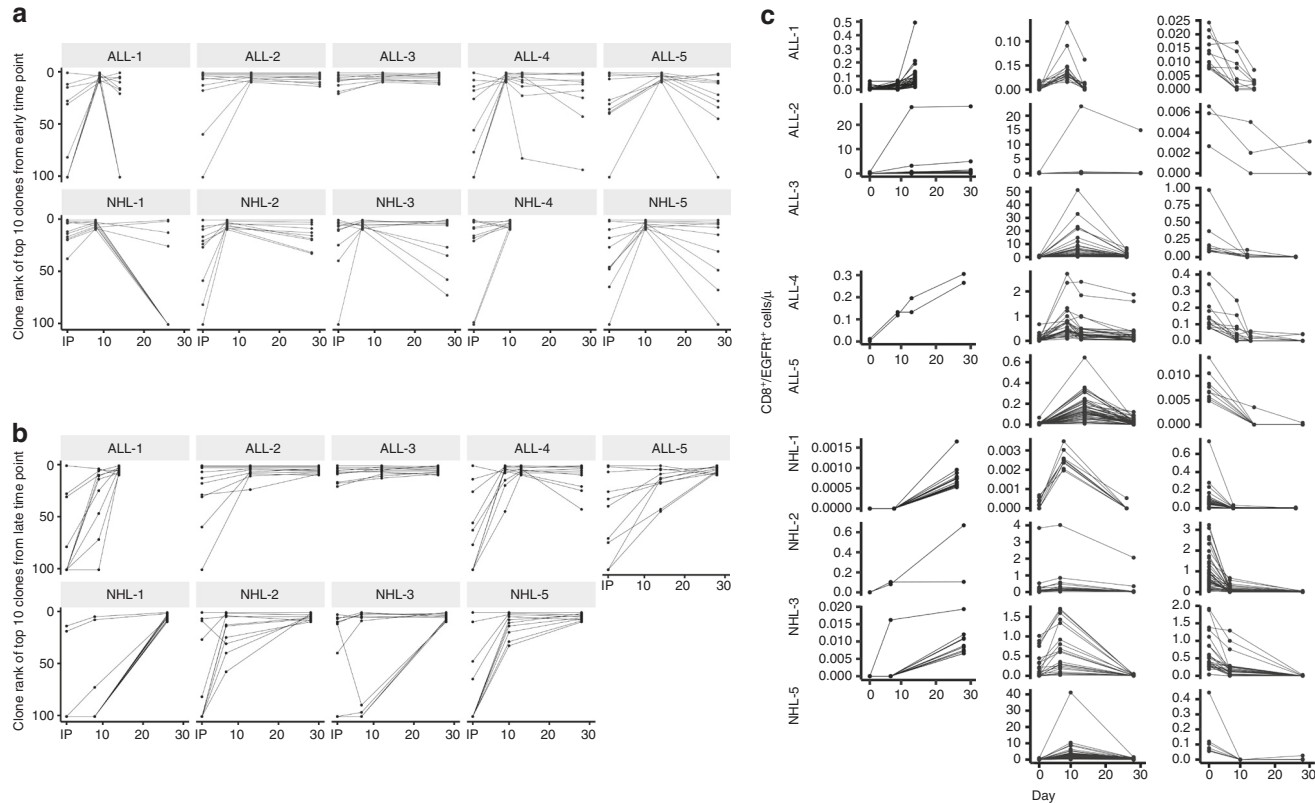

**Fig. 3 Distinct CD8+ CAR-T cell clones exhibit different kinetic behaviors after infusion. a** The rank of the top ten CD8+ CAR-T cell clonotypes from the early time point are shown in the IP and at different time points after infusion. **b** The rank of the top ten CD8+ CAR-T cell clonotypes from the late time point are shown in the IP and at different time points after infusion. All rank positions >100 were designated in the figure as having a rank position of 101. Each line represents an individual clonotype. **c** The absolute counts of the top 30 CD8+ CAR-T cell clones from the IP, early, and late time points were tracked over time. The absolute count at Day 0 represents the absolute count of CD8+ CAR-T cells distributed in blood immediately after infusion, calculated based on the infusion dose. Left, clones whose absolute cell numbers increased in blood after infusion. Middle, clones whose absolute numbers increased in blood early after infusion then decreased. Right, clones whose absolute cell numbers decreased in blood after infusion. There were no increasing clones for ALL-3, ALL-5, and NHL-5. Each line represents an individual clone. Source data are provided as a Source Data file.

CAR-T cell clones after infusion may be associated with changes in gene expression that occur over time during tumor elimination. To study the transcriptional profile of CD8+ CAR-T cells, we selected four additional patients with durable persistence of CAR-T cells following infusion of CD8+ CAR-T cells manufactured from either $T_{CM}$ cells or bulk CD8+ T cells for NHL ($n = 2$) or chronic lymphocytic leukemia (CLL, $n = 2$), respectively (Supplementary Table 1). Using the 10× Genomics platform, we performed scRNA-seq on 62,167 CD8+ CAR-T cells sorted based on truncated human epidermal growth factor receptor (EGFRt) expression from the IP and blood at the early (day 7–14), late (day 26–30), and very late (day 83–112) time points after infusion. In all patients, a majority of the sequenced cells expressed RNA encoding *CD3D*, *CD3E*, *CD8A*, and the single-chain variable fragment (scFv) of the CD19 CAR construct (Supplementary Fig. 3).

Visualization of single-cell transcriptomes by t-distributed stochastic neighbor embedding (t-SNE) revealed that the transcriptional profiles of CD8+ CAR-T cells in the blood at the early, late, and very late time points progressively diverged from that of CAR-T cells isolated from the IPs (Fig. 5a). Differential gene expression (DEG) analysis revealed many differences in gene expression between the IPs and CAR-T cells isolated from blood after infusion (Fig. 5b). CD8+ CAR-T cells in the IPs expressed higher levels of genes associated with the glycolysis pathway, AP-1 transcription factors, and the early

activation marker *CD69*, likely due to in vitro stimulation by CD19+ lymphoblastoid cell line (LCL) during manufacturing (Fig. 5b). Compared with the IP, CAR-T cells at the early and late time points after adoptive transfer expressed higher levels of genes associated with cell-mediated cytotoxicity such as *PRF1*, *GZMB*, and *GZMK*, which declined at the very late time point (Fig. 5b). Gene set enrichment analysis (GSEA) further revealed a transcriptional profile consistent with T-cell activation and an effector response at the early time point that declined through the late and very late time points (Fig. 5c). At later time points, there was a decrease in enrichment of genes associated with the tricarboxylic acid cycle and oxidative phosphorylation, which is consistent with decreasing demand for ATP with declining T-cell activation after tumor clearance. All four patients developed profound B-cell aplasia after CAR-T cell infusion and showed decreasing gene expression of *MKi67* at late and very late time points, consistent with a reduction in CAR-T cell proliferation with depletion of target antigen (Fig. 5d).

We examined the transcriptome data to determine whether there was evidence of exhaustion after infusion. We performed GSEA using two different exhaustion gene sets (Supplementary Table 3). One gene set compares exhausted CD8+ T cells from chronic lymphocytic choriomeningitis virus infection to functional CD8+ T cells after acute infection in mice[24]. The second gene set compares exhausted, tonically signaling GD2.28z CAR-T cells to functional GD2.BBz CAR-T cells generated from healthy

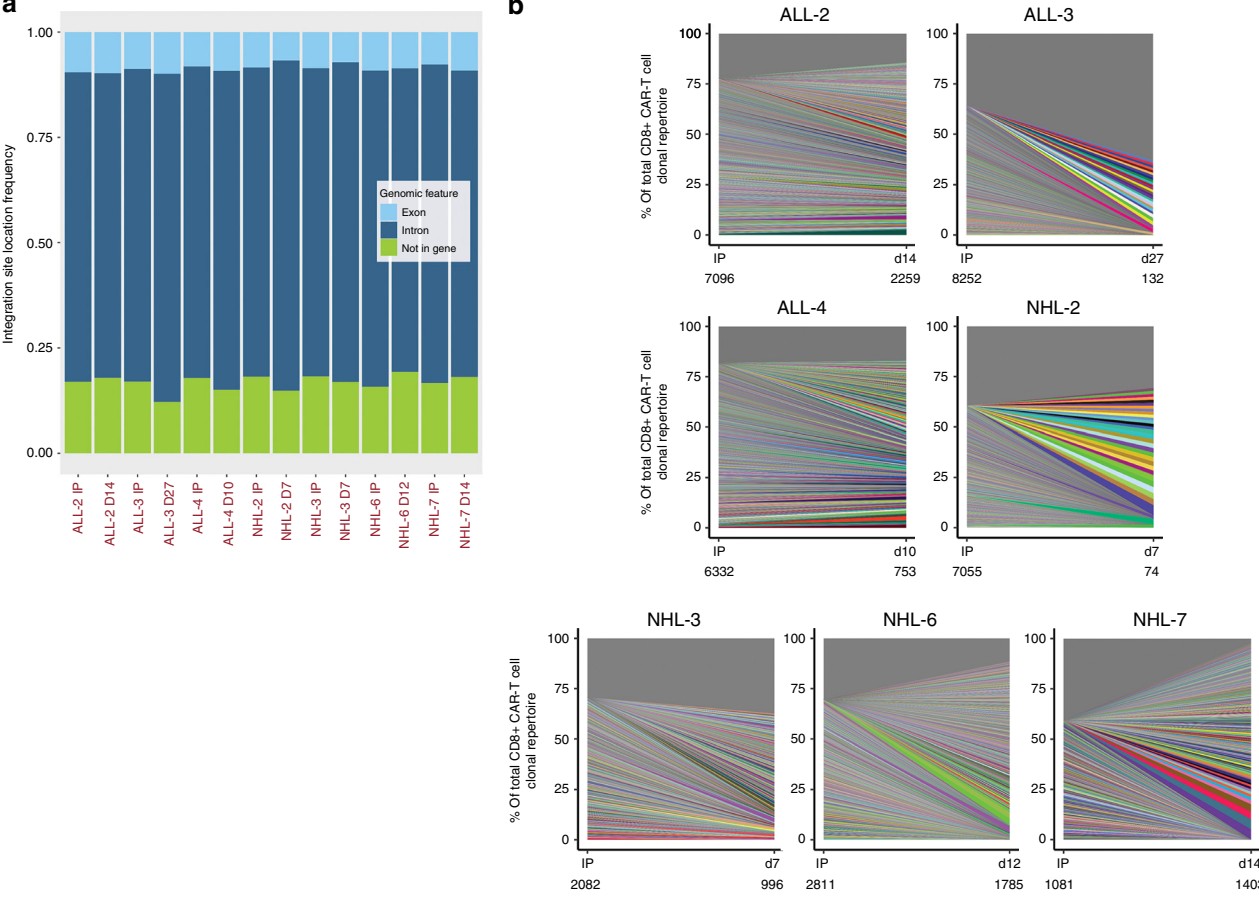

**Fig. 4 Integration site analysis suggests multiple clones contribute to CAR-T cell pools. a** Bar chart depicts the percentage of integration sites found within an exon, intron, or not in a gene for CD8+ CAR-T cells. **b** Graphs represent the contribution (% frequency) of all identified clones by integration site analysis in IP and blood samples collected after infusion (x axis). Each color ribbon represents a unique clone demonstrating ≥1% frequency of sequence reads in a given sample. All other clones are grouped into the gray ribbon at the top of each graph. The total number of unique clones identified in the sample is listed underneath the sample ID for each graph (below the x axis).

human donors[25]. Comparison with the first gene set showed a decrease in expression of the exhaustion gene signature over time after infusion, whereas comparison with the second gene set did not show any changes in expression of the exhaustion gene signature over time (Fig. 5c). Overall, expression of an exhaustion gene signature did not consistently increase over time after infusion. Evaluation of the surface expression of seven inhibitory receptors, including PD-1, LAG-3, TIM-3, KLRG1, TIGIT, 2B4, and CD160, showed that CD8+ CAR-T cells co-expressed higher numbers of inhibitory receptors after infusion, compared with the IP (Fig. 5e). However, expression of multiple inhibitory receptors by CD8+ CAR-T cells remained stable over time after infusion, except in CLL-2, which showed a progressive increase in the fraction of cells co-expressing four or more inhibitory receptors.

**Contribution of infused CAR-T clusters to in vivo repertoire.** We investigated whether transcriptional signatures of CD8+ CAR-T cells in the IP might be associated with different clonal kinetics after infusion. Principal component analysis (PCA) of gene expression residuals demonstrated marked heterogeneity in the transcriptional profiles of CD8+ CAR-T cells in the IP. After infusion, transcriptional heterogeneity of CD8+ CAR-T cells progressively declined in blood over time (Fig. 6a). Unsupervised clustering of the scRNA-seq data further identified four transcriptionally distinct clusters of CAR-T cells that were found in the IP of all four patients (Fig. 6b) and were distinguished by

differential expression of T-cell activation-, cytotoxicity-, mitochondrial-, and cell cycle-associated genes (Fig. 6c). Cells in cluster 2 displayed higher expression of genes associated with cytotoxicity, whereas cells in cluster 4 were distinguished by higher expression of genes associated with proliferation, compared with cells in clusters 1 and 3. When we assigned a cell cycle score to each cell, CAR-T cells in cluster 4 entirely comprised single cells in the S or G2M phase (Supplementary Fig. 4a) confirming that cluster 4 is composed of cycling/proliferating CD8+ CAR-T cells. Pseudotime and RNA velocity analyses of CAR-T cells in the IP did not yield clearly defined trajectories to indicate a developmental relationship between clusters, even after correcting the dataset for cell cycle- and mitochondrial-associated genes with pseudotime analysis (Supplementary Fig. 4b, c).

To link distinct transcriptional programs with clonal expansion and persistence after adoptive transfer, we used paired TCR sequences from the scRNA-seq data to track the clonal behavior of CD8+ CAR-T cells after infusion in NHL-6 and NHL-7. We first identified CD8+ CAR-T cell clonotypes in the IP that had a significantly higher or lower frequency at the early time point, designated as IRF (increased relative frequency) or DRF (decreased relative frequency) clonotypes, respectively (Fig. 7a and Supplementary Table 4)[26]. DEG analysis showed that IRF clones displayed higher expression of *CCL4*, *CD27*, *IFNG*, and cytotoxicity-associated genes (Fig. 7b). Given that IRF clones expressed higher levels of cytotoxicity-associated genes, we

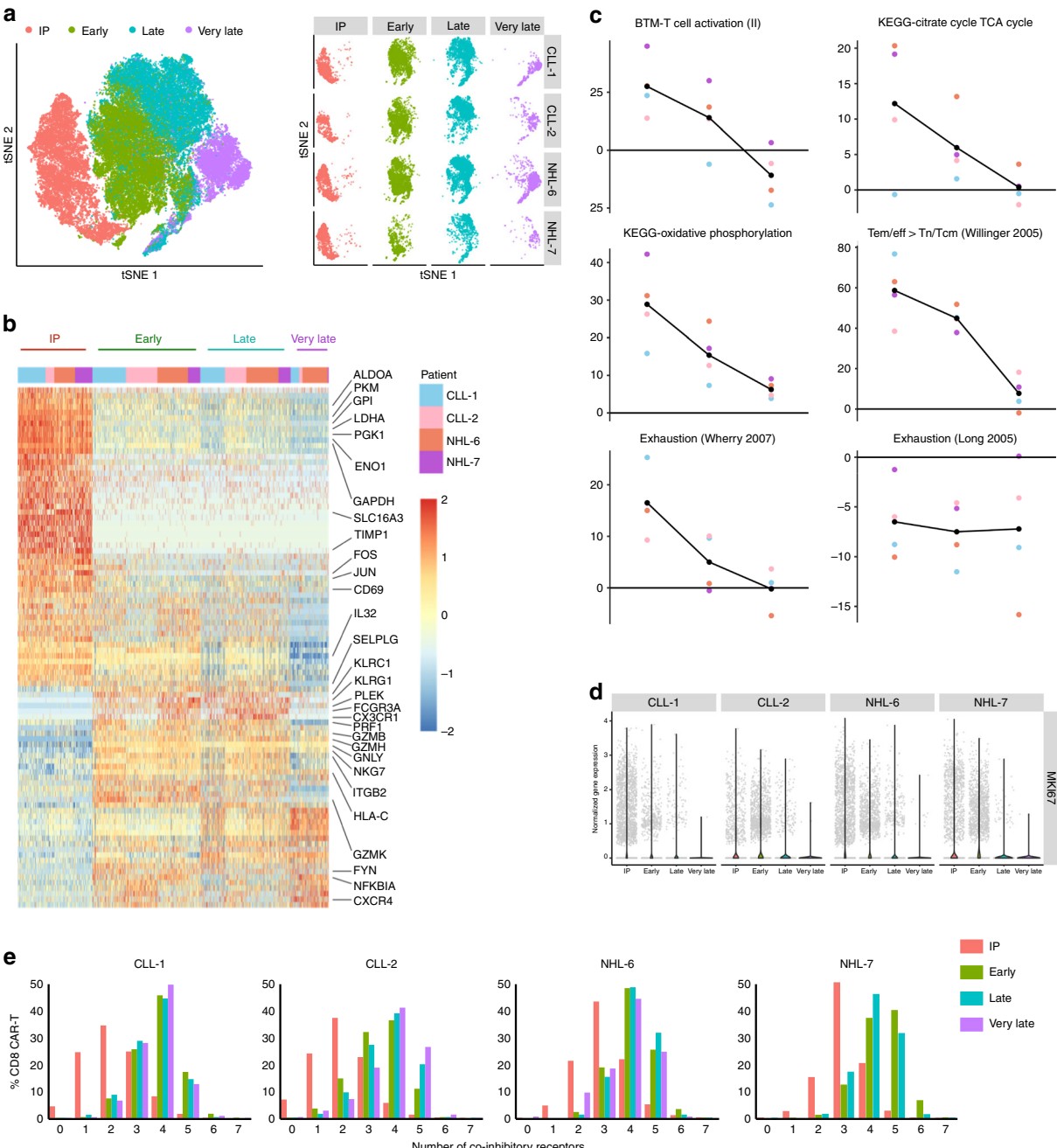

**Fig. 5 Single-cell transcriptome of infused CAR-T cells are distinct from CAR-T cells in blood. a** Left, t-SNE representation of 62,167 CD8$^+$ CAR-T cells concatenated from the IP, early, late, and very late time points of four additional patients. Single cells from the early time point are overlaid on single cells at the late time point. Right, t-SNE analysis of concatenated CD8$^+$ CAR-T cells from each time point in each patient. **b** Heatmap displaying the expression of genes that are differentially expressed between each time point with selected genes highlighted. Only genes that were differentially expressed between time points in all four patients are represented in the heatmap (FDR = 0.05, log2FC = 1.5). Color scale represents gene expression levels as a z-score. **c** GSEA using indicated gene sets was performed on CD8$^+$ CAR-T cells isolated from the IP and blood at distinct time points after adoptive transfer. A combined Z-score was calculated for pairwise comparisons between CD8$^+$ CAR-T cells from each post-infusion sample relative to the IP. A negative Z-score represents higher expression of the gene set in CD8$^+$ CAR-T cells from the IP. Each point represents an individual patient. FDR = 0.01, absolute continuous Z-score > log2(1.5), absolute discrete Z-score > log2(1.5). **d** Violin plot of *MKi67* gene expression at the IP, early, late, and very late time points. **e** Co-expression of zero to seven inhibitory markers, including PD-1, LAG-3, TIM-3, KLRG1, TIGIT, 2B4, and CD160 on CD8$^+$ CAR-T cells from the IP and isolated at the early, late, and very late time points after infusion. Source data underlying Fig. 5e is provided as a Source Data file.

determined the distribution of IRF and DRF clones among the four transcriptionally distinct clusters in the IP. Both IRF and DRF clones were present in all four clusters (Fig. 7c). However, a majority of the IRF clones in both patients were detected in clusters 2 and 4 (NHL-6: 85.6%, NHL-7: 84.6%), which expressed

high levels of genes associated with cytotoxicity and proliferation, respectively (Fig. 7c), despite the finding that these clusters only comprised 42.7% and 27.1% of cells in the IP in NHL-6 and NHL-7 (Fig. 6b), respectively. In contrast, clusters 1 and 3 harbored few IRF clones.

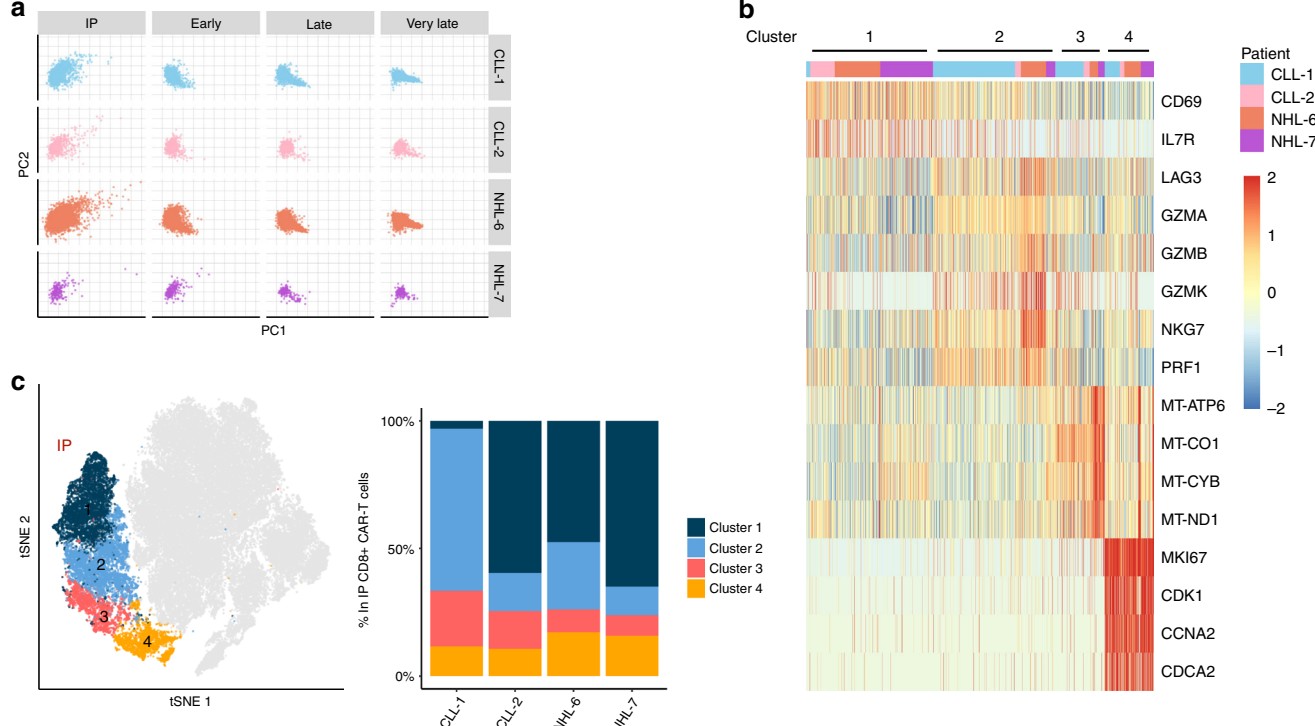

**Fig. 6 CD8$^+$ CAR-T cells in the infusion product form four transcriptionally distinct clusters. a** PCA plot of single-cell gene expression residuals in CD8$^+$ CAR-T cells isolated from the IP and blood at early, late, and very late time points. Gene expression residuals were calculated after adjusting for known sources of heterogeneity: the cellular detection rate, subject, and time point using MAST. Data were downsampled within each patient to account for differences in sample sizes between time points. **b** Unsupervised clustering was performed after concatenation of the scRNA-seq data across all time points and patients. Left, t-SNE plot highlighting the four transcriptionally distinct CD8$^+$ CAR-T cell clusters identified in the IP. Each dot represents a single CD8$^+$ CAR-T cell from the IP (colored) or from the early, late, and very late time points (gray). Right, the percentage of each identified cluster among CD8$^+$ CAR-T cells in the IP is shown for each patient. **c** Heatmap displaying selected genes that are differentially expressed between clusters in the IP. Within each patient, differential gene expression analysis was performed by pairwise comparisons between clusters and only genes that were differentially expressed between clusters in all four patients were represented in the heatmap (FDR = 0.05, log2FC = 1.5). Color scale represents gene expression levels as a *z*-score.

Although IRF clonotypes were identified by their increasing representation at the early time point after infusion, it remained unknown whether IRF clones could persist and contribute to the CD8$^+$ CAR-T cell pool at the late and very late time points. In NHL-6, 100% (29 of 29) of the IRF clonotypes were detected at both the late and very late time points, whereas only 22% (15 of 67) and 9% (6 of 67) of the DRF clonotypes were present at the late and very late time points, respectively (Table 1). In NHL-7, 100% of the IRF clonotypes were detected at the late time point and 53% (10 of 19) were present at the very late time point. In contrast, only 5% (3 of 59) and 3% (2 of 59) of the DRF clonotypes were present at the late and very late time points, respectively (Table 1).

To further determine the contribution of transcriptionally distinct subsets in the IP to the CD8$^+$ CAR-T cell pool after infusion, we identified clonotypes in the IP that were subsequently present or not present at the early, late, or very late time points after infusion. As time after infusion increased, there was a progressive increase in the fraction of persisting clonotypes that originated from cluster 2 in the IP (Fig. 7d). Thus, transcriptionally distinct subsets of infused CAR-T cells differ in their contributions to the CAR-T cell pool in blood after adoptive transfer.

## Discussion

Expansion of CD19-specific CAR-T cells in vivo in response to antigen encounter is a key factor associated with clinical response

and toxicities of CAR-T cell therapy[4,6,11,27]. Previous studies have reported associations between functional attributes in pooled CAR-T cell populations and clinical response[8,28,29], but no studies have examined factors that impact expansion of distinct clones within individual patients. In this study, we examined the in vivo behavior of distinct CAR-T cell clones within individual patients. This approach allowed direct assessment of transcriptional attributes in infused cells associated with clonal expansion in individual patients, regardless of the anti-tumor response, which might be influenced by the tumor burden and tumor microenvironment.

Using TCRB sequencing, we examined the behavior of individual CD8$^+$ CAR-T cells after adoptive transfer in patients that received CD8$^+$ CAR-T cells manufactured from CD8$^+$ T$_{CM}$ cells, to provide greater uniformity in the starting population. As expected, we found that CAR-T cells in the IPs were polyclonal and this polyclonality was maintained in the persisting CAR-T cells in the patients. However, individual clones exhibited different patterns of expansion and contraction in vivo, suggesting that even when starting with a uniform subset of T cells, distinct clones have better capacity to proliferate, survive, and/or remain in the blood after infusion. It remains unclear whether clones that are dominant in the blood are also highly represented at the site of tumor. Additional studies on the CAR-T cell repertoire in the bone marrow and lymph nodes will be important to address the contribution of clones in the IPs to the CAR-T cell pool at the site of the tumor.

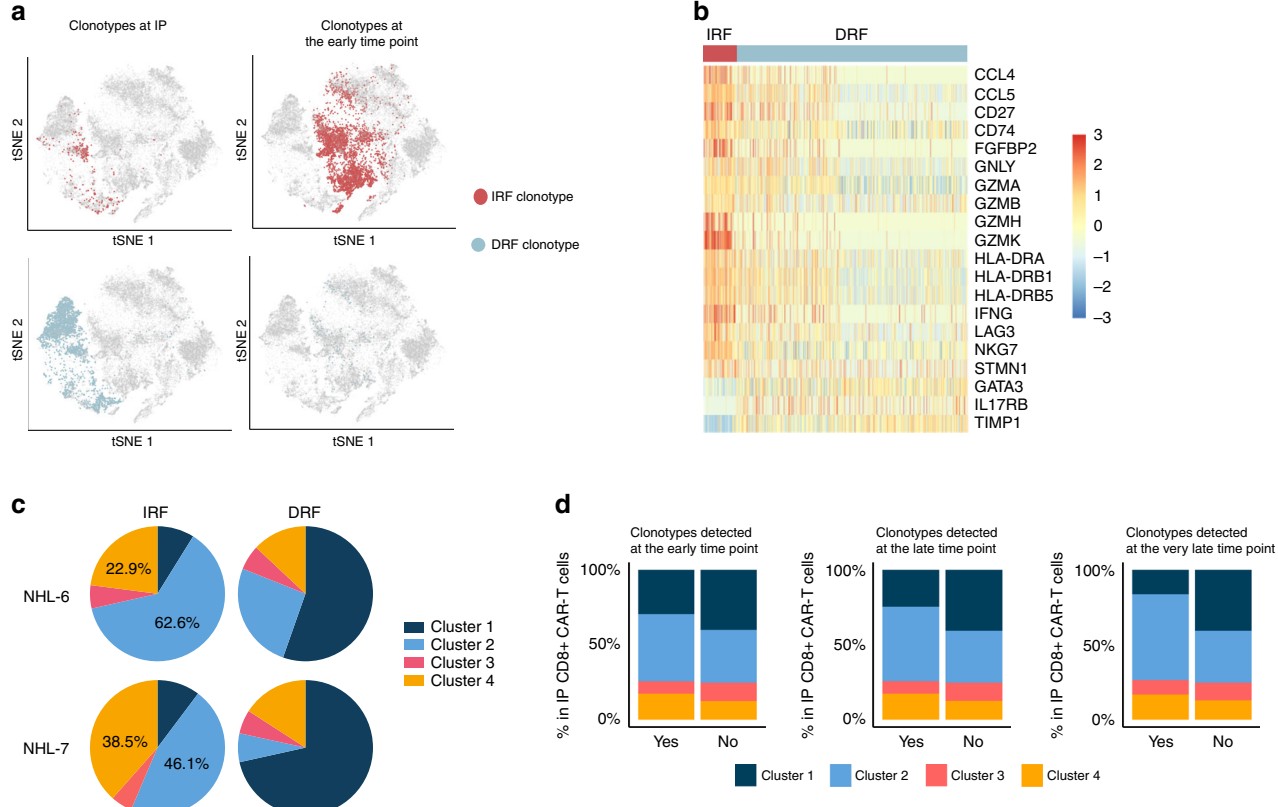

**Fig. 7 Variable contribution of infused CAR-T clusters to in vivo repertoire. a** t-SNE plot showing the localization of IRF clonotypes (red) and DRF clonotypes (blue) in the IP (left) or at the early time point (right). Each dot represents a single CAR-T cell. **b** Heatmap displaying genes that are differentially expressed between IRF and DRF clones in the IP. Color scale represents gene expression levels as a z-score. **c** Pie charts depict the fraction of IRF or DRF clonotypes that are distributed among the four transcriptionally distinct clusters in the IP. The percent of cells in cluster 2 and 4 among IRF clonotypes are reported. **d** Bars depict the fraction of transcriptionally distinct clusters among CD8+ CAR-T cells in the IP that were either detected (Yes) or not detected (No) at the early (Left), late (Middle), or very late (Right) time points after infusion. Source data are provided as a Source Data file.

**Table 1 The proportion of IRF and DRF clones that were present at the late and very late time points after infusion.**

|  | IRF clonotypes | DRF clonotypes |
|---|---|---|
| NHL-6 Late time point | 29/29 (100%) | 15/67 (22%) |
| NHL-6 Very late time point | 29/29 (100%) | 6/67 (9%) |
| NHL-7 Late time point | 19/19 (100%) | 3/59 (5%) |
| NHL-7 Very late time point | 10/19 (53%) | 2/59 (3%) |

scRNA-seq of CD8+ CAR-T cells revealed that the transcriptional profile of CAR-T cells in the blood progressively diverged from that of the IP. Early after infusion, transferred CD8+ CAR-T cells displayed a gene expression profile consistent with activated CD8+ effector T cells, which reflects robust T-cell activation upon antigen encounter. As the tumor and normal CD19+ B cells were eliminated, transcriptional signatures of CAR-T cell activation and proliferation progressively declined without enrichment of an exhaustion gene signature or an increase in co-expression of inhibitory receptors at later times after infusion. Further studies in a larger cohort of patients are required to determine the exhaustion state of CAR-T cells after infusion and whether the single-cell transcriptional profile of CAR-T cells in the blood are representative of CAR-T cells in the tumor.

Single-cell analysis of CD8+ CAR-T cells in the IPs from four patients in whom CAR-T cells were manufactured from CD8+ $T_{CM}$ ($n = 2$) or bulk CD8+ T cells ($n = 2$) revealed the presence of four transcriptionally distinct CAR-T cell clusters. These clusters were observed in all patients, regardless of the phenotype of the starting CD8+ T-cell population used in CAR-T cell manufacturing. The presence of contaminating cells appears unlikely to have influenced the observed transcriptional heterogeneity as the analyzed single cells were sorted to high purity and expressed RNA encoding CD3 subunits, CD8, and the FMC63-derived scFv in the CAR. These clusters were distinguished primarily by the expression of activation-, cytotoxicity-, mitochondrial-, and cell cycle-associated genes. Higher expression of mitochondrial genes in cluster 3 is indicative of pre-apoptotic cells, whereas higher expression of cell cycle-associated genes in cluster 4 signifies cells in the S/G2M phase of the cell cycle. These transcriptionally different cell states may in part be due to stochastic effects of T-cell stimulation and activation during manufacturing, or different levels of CAR expression in individual cells[30]. Studies to evaluate stimulation, transduction, and culture conditions that result in cells with distinct transcriptional attributes may be valuable in optimizing manufacturing strategies.

A number of factors including cell intrinsic properties might contribute to differences in the capacity of CAR-T cells to expand and persist after adoptive transfer. The site of lentiviral vector integration has been reported to affect clonal expansion in which vector integration within the *TET2* gene was found to be the key driver of unique clonal expansion in a single patient[19]. Although

our data cannot exclude the possibility of integration events contributing to clonal kinetics, they do not point to a single vector integration site as the major driver of clonal proliferation in this patient cohort. However, the finding that expanding clonotypes were enriched in integration sites located in genes associated with lymphocyte activation is consistent with robust anti-CD3/CD28-mediated T-cell activation prior to lentiviral transduction being more often associated with subsequent clonal proliferation.

Using paired gene expression and TCR sequences obtained from scRNA-seq data, we tracked the fate of individual CD8$^+$ CAR-T cell clonotypes between the IP and the blood over time after infusion. The majority of IRF clones whose relative abundance increased early after infusion originated from clusters 2 and 4 in the IP, which expressed higher levels of genes associated with cytotoxicity and proliferation. Interestingly, a cytotoxicity gene signature was also highly expressed at the early and late time points relative to the IP, which might be driven by expansion of IRF clones continuing to express this gene signature. IRF clones also displayed higher gene expression of *CD27*, which supports recent findings that the frequency of CD27$^+$ CD8$^+$ T cells in the starting T-cell population and the IP correlates with better clinical outcomes in CLL patients[29]. Importantly, we found that in contrast to DRF clonotypes, a large proportion of IRF clonotypes were also present at the very late time point, suggesting that IRF clones can persist in the blood beyond 3 months after infusion. Additional studies are needed to determine whether IRF clones might also contribute to the CAR-T cell pool at the tumor site and to better understand differences between the transcriptional profiles of CAR-T cells in the tumor compared with blood. Our data reveals that within infused CAR-T cells of defined CD4$^+$:CD8$^+$ composition, CD8$^+$ CAR-T cells in the IP that acquired effector function and proliferated during manufacturing expand and survive in vivo, contributing greatly to the CAR-T cell pool and the anti-tumor response early and late after adoptive transfer. Our findings demonstrate the potential for scRNA-seq to provide unique insights into the in vivo behavior of CAR-T cells after adoptive transfer, which may guide future studies to improve CAR-T cell immunotherapy.

## Methods

**Patient characteristics and treatment regimen**. We studied a subset of adult patients with relapsed and refractory B-cell ALL, NHL, or CLL, who received CD19-specific CAR-T cells in a phase 1 clinical trial (NCT01865617, Supplementary Table 1)[4,6,11] and had sufficient numbers of CD8$^+$ CAR-T cells in the blood after infusion at the early (day 7–14) and late (day 26–30) time points for TCRB repertoire analysis, as well as at the very late (day 83–112) time point for single-cell transcriptome analysis. The study was conducted in accordance with the principles of the Declaration of Helsinki and with the approval of the Fred Hutchinson Cancer Research Center Institutional Review Board. CD19-specific CAR-T cells were manufactured from isolated bulk CD4$^+$ T cells and either T$_{CM}$ or bulk CD8$^+$ T cells, as previously described[4,6,11]. In brief, T-cell subsets were isolated from leukapheresis products, stimulated with paramagnetic beads (CTS Dynabeads CD3/CD28), and then transduced with a lentivirus incorporating a transgene encoding a CD19-targeting CAR containing a 4-1BB/CD3ζ signaling domain and an EGFRt, which enabled identification of transduced cells by flow cytometry (Supplementary Fig. 5a)[31]. CAR-T cells were stimulated during culture with an irradiated, allogeneic CD19$^+$ LCL cell line and were formulated 9–11 days after LCL stimulation in a 1:1 ratio of CD4$^+$:CD8$^+$ CAR-T cells for infusion[4,6,11].

**CAR-T cell isolation, enumeration, and immunophenotyping**. CD8$^+$ CAR-T cells for TCRB sequencing or scRNA-seq were isolated from cryopreserved aliquots of the IP or peripheral blood mononuclear cell (PBMC), washed in RPMI containing 50 U/ml of benzonase (Novagen), incubated with Live/Dead Fixable Violet stain, then stained with biotinylated anti-EGFRt monoclonal antibody (cetuximab), followed by fluorochrome-conjugated streptavidin and antibodies to CD4 and CD8. CD8$^+$ CAR-T cells were isolated as CD8$^+$/CD4$^-$/EGFRt$^+$ events in a singlet lymphocyte forward-scatter (FS)/side-scatter (SS) gate using a BD Aria 2 flow sorter (Supplementary Fig. 5b).

The absolute CD8$^+$ CAR-T cell count in blood was determined by multiplying the percentage of CD8$^+$ CAR-T cells in a viable CD45$^+$ lymphocyte FS/SS gate by the absolute lymphocyte count determined from a complete blood count

performed on the same day. The absolute count of CD8$^+$ CAR-T cells distributed in blood immediately after infusion was determined by dividing the number of infused CD8$^+$ CAR-T cells by the estimated total blood volume of the recipient (BloodVol = Weight × AvgBloodVol; AvgBloodVol for women = 65 ml/kg and men = 75 ml/kg).

For analysis of the surface immunophenotype of CAR-T cells, aliquots of the IPs and PBMCs were stained with Live/Dead Fixable Blue stain followed by fluorochrome-conjugated antibodies to EGFRt, CD45, CD3, CD4, CD8, LAG-3, PD-1, TIM-3, KLRG1, TIGIT, 2B4, and CD160 with analysis using a BD FACSymphony flow cytometer.

**TCRB sequencing**. Sequencing of the TCRB CDR3 region in CD8$^+$ CAR-T cells was performed by Adaptive Biotechnologies, with subsequent analysis using R. To compare TCRB diversity between samples of different sizes, we downsampled CDR3 sequences to the minimum number of productive sequences, downsampling 1000 times for each sample[22]. For each downsampled dataset, we calculated the Shannon entropy index and the median Shannon entropy was used for comparisons between samples. The relative frequency of a CD8$^+$ CAR-T cell clonotype in a sample was defined as the fraction of the total CD8$^+$ CAR-T cell TCRB sequences that was occupied by the clonotype. The absolute count of an individual CAR-T cell clone in blood was determined by multiplying the relative frequency of the clonotype by the absolute CAR-T cell count.

**Integration site analysis**. Integration site analysis (ISA) was performed on CD8$^+$ T cells isolated from the IP and from blood after adoptive transfer. Processing of gDNA to amplify integration loci included MGS-PCR methods[32]. Briefly, amplification of integration loci occurs through two PCR reactions. In the first PCR, the forward primer is aligned to the lentivirus long terminal repeat (LTR) and the reverse primer is aligned to a synthetic linker oligonucleotide that was assembled and ligated to fragmented genomic DNA. The forward primer (LTR) in the second PCR is nested and aligns to the 3′-end of the sequence amplified by the LTR primer used in the first PCR. The reverse primer for the linker cassette is the same for both reactions. Primer sequences are listed in Supplementary Table 5.

ISA reads were sequenced using the Illumina Miseq next-generation sequencing platform for paired-end reads. A detailed list of samples is included in the dataset available for download (link to be provided at time of publication). For Illumina data, the forward and reverse reads were stitched using PEAR with the *-q 30* option to trim sequence reads after two bases with a quality score below 30 were observed[33]. Stitched FASTQ files and raw FASTA files for all sequencing data were filtered using a custom c++ program. Each read was compared with the reference provirus LTR sequence. Reads with <90% match to LTR sequence were discarded. The LTR sequence was trimmed off of remaining reads. Reads were then compared with vector sequence (as opposed to genomic insertion sequences). Reads with ≥80% match to vector sequence were discarded. Remaining reads were output in FASTA format for alignment.

Identified genomic fragments were aligned to the human Genome Reference Consortium Human Build 38 (GRCh38) [RefSeq Assembly: GCF_000001405.26], to determine the chromosome, locus, and orientation of integration (e.g., Chr14_8020175_ + ). GRCh38 was downloaded from the UCSC genome browser (http://genome.ucsc.edu/)[34]: filtered and trimmed sequence reads were aligned to the reference genome using BLAT with options -out = blast8, -tileSize = 11, -stepSize = 5, and -ooc = hg11-2253.ooc.[35]

The hg11-2253.ooc file contains a list of 11-mers occurring at least 2253 times in the genome to be masked by BLAT and was generated using the following command:

$blat hg38.2 bit /dev/null /dev/null -tileSize = 11 -stepSize = 5 -makeOoc = hg11-2253.ooc -repMatch = 2253

as recommended by UCSC [http://genome.ucsc.edu/goldenpath/help/blatSpec.html[35] and http://genome.ucsc.edu/FAQ/FAQblat.html#blat6]. Resulting blast8 files were parsed using a custom python script. Any read with BLAT alignment length <30 or alignment start >10 is discarded. The top scoring alignment and all alignments within 95% of the top score were saved for each read. The ratio of the best alignment to the second-best alignment is the degree with which the insert can be mapped to one location in the genome (multi-align-ratio). Starting with the highest count reads, reads with matching alignments were combined. Reads with multiple possible alignments were not discarded at this point but were grouped together with other reads with the same alignment(s). For each group of reads with matching alignments, the original FASTA sequence files were read, then the sequences were aligned with Clustal Omega[36,37]. This alignment was used to build a single consensus sequence for the alignment group. The consensus sequence was then used to search the pool of all sequences that could not be aligned by BLAT and any sequence with ≥90% identity was merged into the group. Starting with the highest count groups and continuing with the lowest count groups, groups with ≥90% sequence similarity were merged. Finally, when comparing all sequence files for one test subject, all groups with exact alignment matches were merged into one clone ID. Clone IDs with exact consensus sequence matches were also merged. Non-uniquely aligned groups (multi-align-ratio ≥ 0.9) that had ≥90% similar reference sequences were also merged into a single Clone ID[38].

ISA clone contributions were calculated by dividing the number of sequence reads for each clone by the total number of sequence reads associated with any

clone ID for the sample. The mean maximum frequency of clones was calculated. A data frame was constructed including three columns: (1) days after treatment (or IPs); (2) clone frequency in the sample; and (3) clone identifier (genomic locus). The data frame was visualized using the R package *ggplot2* (http://ggplot2.org/).

We then used normalization to convert the number of sequence reads associated with each clone to frequencies for each sample. At this point, we determined the maximum contribution frequency of each clone observed. The contribution frequencies for each of these clones at each sampling time point is recorded. Each clone has two entries in the data frame, one for each sampling time point, with 0% values entered where sequence reads were not observed. This is repeated for every clone identified in any sample from the same subject. Visualization is then done using *ggplot2* (http://ggplot2.org/) using *geom_area/geom_ribbon*. Clones contributing ≥1% of the detected pool in any sample from the same subject are denoted by a color ribbon. All remaining identified clones are grouped into the gray ribbon at the top of each plot.

Integration sites that increased in relative abundance by at least fivefold between the IPs and after infusion were identified and combined across all patients. Gene Ontology (GO) enrichment analysis was performed on this list of expanding integration sites using the Gene Ontology Consortium website (http://geneontology.org/) to find significantly enriched pathways[39–41]. This analysis was repeated for integration sites that decreased in relative abundance by at least fivefold between the IP and after infusion. Statistics were performed by the GO consortium website using false discovery rate (FDR) and corrected *p*-values associate with each pathway was represented in a heatmap.

**Transcriptional profiling of single CD8+ CAR-T cells.** Construction of scRNA-seq libraries was performed using the Chromium Single Cell 5′-library and V(D)J enrichment kit as per manufacturer's instructions (10× Genomics). For each sample, a 5′-gene expression library and an enriched V(D)J library was constructed and sequenced using the HiSeq 2500 platform (Illumina). The Cell Ranger Single-Cell Software Suite (version 2.1.0) was used to perform sample demultiplexing, barcode processing, and single-cell gene counting. Raw base BCL files were demultiplexed using the Cell Ranger *mkfastq* pipeline into sample-specific FASTQ files. FASTQ files were processed individually using the Cell Ranger *count* pipeline, which used the STAR software[42] to align reads to the pre-built GRCh38 human reference genome provided by Cell Ranger (10× Genomics). Aligned reads were then filtered for valid cell barcodes and unique molecular identifiers (UMIs). Cell barcodes 1 Hamming distance away from a list of known barcodes were considered. UMIs with a sequencing quality score >10% without homopolymers were retained as valid UMIs. A UMI with 1 Hamming distance from another UMI with more reads, for the same gene and same cell, was corrected to the UMI with more reads. All samples were aggregated using the Cell Ranger *aggr* pipeline resulting in one gene-barcode count matrix to be used for downstream analyses. A correction for sequencing depth was performed during the aggregation[43].

Following sequence alignment and filtering, data from cells with unique gene counts <200, a percentage of mitochondrial genes >20% or >40,000 UMIs were removed. Based on these criteria, 2340 cells were discarded, resulting in 62,167 CD8+ single CAR-T cells for analysis. Standard analyses were performed using the Seurat R package[44]. Only genes with at least one UMI count detected in at least three cells were used. A library-size normalization was performed for each cell. UMI counts were scaled by the total expression in each cell and multiplied by 10,000. The data were then log-transformed and corrected for unwanted sources of variation such as the number of detected UMIs, percentage of mitochondrial gene content, and different patient origin using the *ScaleData* R function as described in the Seurat manual[44]. The corrected-normalized gene-barcode matrix was used to perform PCA, clustering, and t-SNE analyses, whereas the normalized gene-cell barcode matrix was used for the MAST analysis as described below.

The top 1000 most variable genes were kept as inputs to compute the PCA. The top 15 principal components were then down-selected for t-SNE visualization and clustering. One thousand iterations of the t-SNE Barnes-hut implementation using a perplexity value of 30 were performed. Cell clustering was performed using a graph-based clustering method implemented in Seurat (*FindClusters* R function—share nearest neighbor modularity optimization-based clustering algorithm) using default parameters.

DEG analysis and GSEA analysis were performed using the MAST R package[45]. Within each patient, a logistic regression model was used to test the DEG rate between groups, while a Gaussian generalized linear model described expression conditionally on non-zero expression estimates. The model was also corrected for the cellular detection rate, defined as the proportion of expressed genes in a given cell. Genes were declared significantly differentially expressed at a FDR of 5% and absolute log2 fold change > log2(1.5). Gene sets were declared significant at an FDR of 1%, absolute continuous *Z*-score > log2(2.5) and absolute discrete *Z*-score > log2 (2.5). Blood transcriptome modules[46] and Kyoto Encyclopedia of Genes and Genomes (KEGG)- and T-cell-related pathways[24,25,47] (Supplementary Table 4) were used as gene sets. KEGG gene sets were downloaded from MSigDB database[48].

The CellCycleScoring R function from the Seurat R package was used to assign a cell cycle score to each cell. This function calculates S and G2M scores and predicts classification of each cell in either S, G2M, or the G1 phase.

**Single-cell trajectory analysis.** The latest version of the Monocle 3R package was used to construct single-cell trajectories of cells from clusters in the IPs[49–51]. The first step in the Monocle 3 workflow (https://cole-trapnell-lab.github.io/monocle3/docs/trajectories/) was to normalize and pre-process the data. Using the *align_cds* R function, patient ID was selected as the alignment group and the data were corrected for cell cycle scores, the number of detected UMIs, and the percentage of mitochondrial genes. Next, the dimensionality of the data is reduced using uniform manifold approximation and projection with the *reduce_dimension* R function. Trajectory inference analysis was performed using the *cluster_cells* and *learn_graph* R functions using the default settings.

For each patient, the dropEst pipeline was used to generate count matrices that calculate the fraction of intronic and exonic UMIs separately[52]. The 10× bam files were used as inputs with the following options -V -C 10000 -m -f. Separate .rds files containing intronic, exonic, or exon/intron spanning matrices were produced and used as inputs for RNA velocity analysis. The velocyto R package was then used to analyze the expression dynamics of cells in the IPs[53]. Cells from clusters 1, 2, 3, and 4 representing the IPs were used as inputs. After filtering genes using the *filter.genes.by.cluster.expression* R function for both intronic and exonic matrices, estimation of RNA velocity was performed using the *gene.relative.velocity.estimates* R function with kCells of 25, fit.quantile of 0.02, and PCA projections from the previous Seurat analysis as proposed in the online tutorial for the cell.dist parameter. RNA velocities were then visualized on our existing embedding (t-SNE) using the *show.velocity.on.embedding.cor* R function.

**TCRB sequencing of single CD8+ CAR-T cells.** Generation of TCRB sequences and annotations in single CAR-T cells was performed on scRNA-seq data by Cell Ranger *vdj*. Cell Ranger *vdj* takes as inputs FASTQ files previously produced from Cell Ranger *mkfastq* and performs V(D)J sequence assembly and paired clonotype calling. TCRB sequence analysis was performed using R. The input data consisted of the observed number and fraction of each TCRB sequence in each sample. We defined IRF and DRF clonotypes as those with higher or lower relative TCRB clonotype abundance, respectively, in CAR-T cells at the early time point compared with the IPs (Fisher's exact test, FDR of 5%).

**Reporting summary.** Further information on research design is available in the Nature Research Reporting Summary linked to this article.

## Data availability

TCR-seq data are provided as a Source Data file. All sequence data obtained in the integration site analysis study are available at National Center for Biotechnology Information Sequence Read Archive (NCBI SRA) under accession number PRJNA589633. scRNA-seq data are available at National Center for Biotechnology Information Gene Expression Omnibus (NCBI GEO) under accession number GSE125881. The source data underlying Figs. 1a, c, d, 2a, b, 3a-c, 5e, and 7c, and Supplementary Figs. 1 and 2 are provided as a Source Data file.

## Code availability

All custom R and Python codes are available on request or at https://github.com/ValentinVoillet/CAR-T. All other codes are publicly available and cited in the appropriate methods description.

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

## Acknowledgements

We thank the staff of the Fred Hutchinson Cancer Research Center (FHCRC) Cell Processing Facility, Seattle Cancer Care Alliance (SCCA) Cell Therapy Laboratory, the FHCRC Integrated Immunotherapy Research Center (IIRC), and the SCCA Bezos Family Immunotherapy Clinic. This work was supported by Juno Therapeutics/Celgene, Inc., IIRC pilot grant, the Bezos Family, and the NHLBI funded National Gene Vector Biorepository at Indiana University (Contract # 75N92019D00018).

## Author contributions

A.S. designed and performed experiments, analyzed data, and wrote the manuscript. V.V. analyzed the scRNA-seq data and wrote the manuscript. L.A.H. designed and performed the TCRB sequencing experiments, analyzed data, and wrote the manuscript. H.D. analyzed TCRB sequencing experiments and contributed to the preparation of the manuscript. M.Y., R.H., and V.G. performed experiments and/or analyzed data. M.E.W., D.P., M.R.E., J.E.A., and H.-P.K. performed and/or analyzed integration site analysis data. S.R.R., D.G.M., R.G., and P.S.L. contributed to the discussion of the project results and provided manuscript feedback. C.J.T. supervised the project and contributed to the overall analysis and writing of the paper, and all authors reviewed the final version of the manuscript.

## Competing interests

S.R.R. received research funding from Juno Therapeutics, a Celgene company, has patents licensed to Juno Therapeutics, a Celgene company, has equity ownership in Celgene, and has served on advisory boards for Adaptive Biotechnologies, Cell Medica, Juno Therapeutics, a Celgene company, and NOHLA. D.G.M. received research funding from GlaxoSmithKline and Juno Therapeutics, a Celgene company. H.P.K. is a consultant to and has ownership interests with Rocket Pharma and Homology Medicines, is a consultant to CSL Behring and Magenta Therapeutics, and is an inventor on patent applications (#62/351,761, #62/428,994, and #PCT/US2017/037967) submitted by the Fred Hutchinson Cancer Research Center that cover the selection and use of cell populations for research and therapeutic purposes, as well as strategies to assess and/or produce cell populations with predictive engraftment potential. C.J.T. received research funding from Juno Therapeutics, a Celgene company, and Nektar Therapeutics, has patents licensed to Juno Therapeutics, a Celgene company, has served on advisory boards, has equity ownership in Caribou Biosciences, Eureka Therapeutics, and Precision Biosciences, and has served on advisory boards for Aptevo, Juno Therapeutics, a Celgene company, Kite, a Gilead Company, Nektar Therapeutics, Novartis, Allogene, Myeloid Therapeutics, and PACT Pharma. R.G. has received consulting income from Juno Therapeutics, Takeda, Infotech Soft, Celgene, has received research support from Janssen Pharmaceuticals and Juno Therapeutics, and declares ownership in Cellspace Biosciences. The remaining authors declare no competing financial interests. Companies funding this research did not have any role in the study design or data analysis and interpretation.
