## [Peer Review File · Nature Communications]

Reviewers' comments:

Reviewer #1, expertise in TCR repertoire in immunotherapy (Remarks to the Author):

Using single-cell RNA sequencing and TCRB gene sequencing, Alyssa Sheih and colleagues have presented here single-cell transcriptional profiling of adoptively transferred CD19-specific CD8+ CAR-T cells along with its clonal kinetics. The major findings of the report include that the infused CAR-T cells were polyclonal and they remained as such several days after infusion. Four transcriptionally distinct clusters were identified in the scRNA-seq data from 4 patients and most of the clones that persisted to very late timepoints originated from 2 of these clusters that were enriched in cytotoxicity and proliferation genes. This is a well-written and easy to understand manuscript that will lead to a better understanding of the in vivo behavior of infused CAR-T cells. However, the following revisions will further strengthen the paper.

1. In supplementary table 2, the total number of TCRB gene sequences are consistently lower after infusion as compared to IP. Is this due to differences in sequencing depth of the samples or due to the nature of the cells after infusion? Please explain.
2. For figure 3a on page 5, you have reported "Interestingly, the top ten clonotypes from the early time point generally remained among the top ten clonotypes at the late time point in most patients, except in NHL-1 and NHL-3". Please considering adding NHL-5 to this list as the clonotypes seem to behave in the same manner as in NHL-1 & NHL-3.
3. In figure 4c, your data suggests based on Wherry 2007 exhaustion gene set, that the expression of exhaustion genes at early time point after infusion is higher than infusion product (IP) and it decreases progressively over time. This is inconsistent with the Long 2015 gene set, which suggests that the IP is more exhausted as compared to all the timepoints after infusion (as assessed from the negative Z scores). Moreover, both datasets are then in inconsistent with the findings in Fig. 4e, which suggests that cells are more exhausted after infusion, but the level of exhaustion does not decrease over time as suggested from the GSEA using Wherry 2007 gene set. Please explain this inconsistency in the discussion and possibly consider providing a list of gene sets in the supplementary.
4. In Fig. 5a, PCA analysis was provided only for cells that declined in blood over time after infusion. Please also consider doing the same for cells that progressively increased.
5. On page 9, paragraph 2, please add references to Table 1 in the text.

Reviewer #2, clinical CAR T cell expert (Remarks to the Author):

Sheih et al analyzed the clonal selection in vivo of CD19.CAR-T cells after adoptive transfer and performed single-cell transcriptional profile of CAR-T cells. The authors focused on CD8+ T cells. The manuscript is an extension of previous observations published by the same group. Methodologically the manuscript is in general accurate, however the overall impact of the reported data is rather limited.

Major comments

- 1) The authors observed a contraction of the TCR beta repertoire of the CAR-T cells in samples collected at later time points as compared to early time points, at the peak of CAR-T cell expansion in vivo, which is largely expected. It remains unclear if the analysis of the TCR repertoire can help in identifying which are the characteristics of the CAR-T clones that are expanding/persisting better in vivo. It may be possible, that in addition to specific functional characteristics of the T cells expressing the CAR (which are not identified in this manuscript), the clonal insertion of the vector may play a relevant role. This aspect should be experimentally analyzed.
- 2) The performed analysis in samples of peripheral blood may not reflect what occurs in the bone marrow and lymphonodes. Are CAR-T cells in the peripheral blood, bone marrow and lymphonodes showing similar TCR repertoire at different time points?
- 3) The single-cell transcriptional profile shows that CAR-Ts are highly cytotoxic at early time points

and became more “transcriptionally quiescent” at later time points during the contraction phase of CAR-T cells. This is largely expected and not particularly informative. It would be more relevant to assess the single-cell transcriptional profile of CAR-T cells at later time point in bone marrow, where these cells may remain more active since they continue eliminating normal B cells that the bone marrow is regenerating.

4) The single-cell transcriptional profile includes 2 patients with NHL and 2 patients with CLL. These are really different diseases in term of “T cell quality” as recently reported by University of Pennsylvania. Furthermore, CAR-T cells for NHL and CLL were manufactured using two completely different approaches (TCM versus bulk T cells).

Reviewer #3, translational and clinical CAR T cell expert (Remarks to the Author):

Sheih and colleagues have conducted a comprehensive analysis of the persisting CD8 CD19 CAR T cells in patients with heme malignancies. The work is done using state of the art single cell profiling of the CAR T cells captured during manufacturing and at various times following infusion in patients. There are a number of inherent biases in this type of study, since it only looks at successful treatments, rather than in patients that have poor engraftment and no clinical responses. There is an implied bias/assumption that the persisting cells have preferred transcriptional states (i.e. cluster 2 and 4) and that the “loser” CARs have a cluster 1 signature. It is possible that other factors might be involved such as cell extrinsic mechanisms such as immune responses arising to the CAR that cause rejection and a “loser” state. In addition, authors expand the CAR T during manufacturing with an EBV LCL line (?autologous), and don’t state that host responses directed to CAR T with TCRs reactive to EBV antigens might be involved in persistence, independent of the transcriptional profiling. Authors should comment on this. In addition, please address:

1. Figure 1d. Patient ALL-2 has a dominant clone at the late timepoint, also shown in Supple Fig 1. Where is the integrated CAR and is it near a gene that might be implicated in clonal T cell proliferation? Same question for patient NHL-2 clonotype who has emergence of clonal outgrowth.

2. Related to the above, authors previously reported that the peak CAR T in patients was polyclonal (Turtle, JCI 2016). This doesn’t seem consistent with the oligoclonal outgrowths in patients ALL-2 and NHL-2.

3. Figure 3c. why are only 6 patients shown in this panel? For example, why isn’t data from patient ALL-2 shown in this panel?

4. Page 15. “Following sequence alignment and filtering, data from cells with unique gene counts less than 200, a percentage of mitochondrial genes greater than 20% or greater than 40,000 UMIs were removed, resulting in 62,167 CD8+ single CAR-T cells for analysis.”

Authors should state how many cells were discarded at this step so that the reader can judge how many cells were required to obtain 62,167 CD8 CAR T cells.

5. Supple Fig 2. How do authors explain the decreased scFv CAR expression compared to CD3 and CD8A? At the protein level, there are approximately equal numbers of CAR and TCR/CD3 reads. Is silencing of the transgene occurring?

Reviewer #4, expertise in T cell repertoire in immunotherapy (Remarks to the Author):

This study utilizes TCR beta chain sequencing to investigate the expansion and maintenance of anti-CD19 CAR T-cell clonotypes after transfer to patients with B-cell malignancies. Therefore, prevalence

of TCR beta clonotypes is measured in the infusion product (IP) and at later time points in the patients receiving these products. In addition, CAR T-cell populations are subjected to single-cell RNA sequencing (scRNAseq) (4 populations sequenced) and further TCR beta sequencing (2 of the 4 populations) before and after infusion into patients. These datasets are then used to identify clonotypes that show increased (IRF) and decreased relative frequency (DRF) in patients vs. IPs. Finally, a set of differentially expressed genes in IRF clones vs. DRF clones is identified. The main findings of the manuscript are 1) that clonotypes differ substantially in their expansion after infusion 2) that increased expansion after infusion correlates with differential gene expression already in the IP.

The technical effort made and the technical quality of the applied approaches is high—especially for a clinical setting. The basic observation, of changing clonotypic dominance is surprising and at least to my knowledge novel. Still, I feel that the manuscript, in its current form, lacks conceptual advance.

Major concerns:

To warrant publication in Nature Communications, I feel that two sets of questions would have to be addressed:

- 1) How do the distinct transcriptional clusters identified via scRNAseq in the IP align developmentally? Is cluster 2 a precursor of cluster 1 or is it the other way around? Gaining more insight here, e.g. via pseudotime analyses, could lay the foundation for answering the second set of questions:
- 2) Can the clusters/subsets found in the IP be preferentially induced by modulating the production process of CAR T cells? Do they preferentially arise at certain time points of the production process?

Answering question 1) will require trajectory inference methods applied to the existing scRNAseq data, answering question 2) will require additional “wet lab” experiments coupled to (sc)RNAseq and/or measurement of key transcripts/molecules identified in the differential gene expression analysis.

Minor concerns:

Further on, it may be helpful to limit the differential gene expression analysis to IRF and DRF clonotypes showing particularly distinct expansion behaviors (e.g. the three strongest expanding vs. the three strongest decreasing clones).

On a technical note: Cluster 4, showing high Ki67, CDK signatures, should not be defined or treated as a subset but simply as single cells identified by chance in the S or G2M phase. The authors are correct in stating that overrepresentation of certain clonotypes in this cluster is indicative of higher proliferation activity. But to efficiently perform pseudotime and detailed clustering analyses the authors may profit from correcting their dataset for the proliferation associated artefacts that generate cluster 4. Further on, Cluster 3, showing strong mitochondrial RNA signatures likely consists of (pre-)apoptotic cells. This cluster may as well disturb detailed cluster and pseudotime analyses.

Response to Reviewer 1 Comments

Reviewer #1, expertise in TCR repertoire in immunotherapy (Remarks to the Author):

Using single-cell RNA sequencing and TCRB gene sequencing, Alyssa Sheih and colleagues have presented here single-cell transcriptional profiling of adoptively transferred CD19-specific CD8+ CAR-T cells along with its clonal kinetics. The major findings of the report include that the infused CAR-T cells were polyclonal and they remained as such several days after infusion. Four transcriptionally distinct clusters were identified in the scRNA-seq data from 4 patients and most of the clones that persisted to very late timepoints originated from 2 of these clusters that were enriched in cytotoxicity and proliferation genes.

Comment 1: This is a well-written and easy to understand manuscript that will lead to a better understanding of the in vivo behavior of infused CAR-T cells. However, the following revisions will further strengthen the paper.

Response: We appreciate this positive feedback from the reviewer that our manuscript will lead to a better understanding of the in vivo behavior of infused CAR-T cells. We have made revisions, as suggested, to further strengthen the paper.

Comment 2: In supplementary table 2, the total number of TCRB gene sequences are consistently lower after infusion as compared to IP. Is this due to differences in sequencing depth of the samples or due to the nature of the cells after infusion? Please explain.

Response: The total number of TCRB gene sequences was lower after infusion because fewer CAR-T cells could be isolated from blood for analysis at the early and late time points compared to the infusion product. To account for these differences in the total number of TCRB gene sequences between time points, we downsampled CDR3 sequences to the minimum number of productive sequences before comparing clonal diversity. This analysis is further described on page 16, line 386-390 of the manuscript as follows:

“To compare TCRB diversity between samples of different sizes, we downsampled CDR3 sequences to the minimum number of productive sequences, downsampling 1000 times for each sample²². For each downsampled dataset, we calculated the Shannon entropy index and the median Shannon entropy was used for comparisons between samples.”

Comment 3: For figure 3a on page 5, you have reported “Interestingly, the top ten clonotypes from the early time point generally remained among the top ten clonotypes at the late time point in most patients, except in NHL-1 and NHL-3”. Please considering adding NHL-5 to this list as the clonotypes seem to behave in the same manner as in NHL-1 & NHL-3.

Response: Thank you for this suggestion. We agree that NHL-5 behaves in the same manner as NHL-1 and NHL-3 and have modified the manuscript on page 6, line 120.

Comment 4: In figure 4c, your data suggests based on Wherry 2007 exhaustion gene set, that the expression of exhaustion genes at early time point after infusion is higher than infusion product (IP) and it decreases progressively over time. This is inconsistent with the Long 2015 gene set, which suggests that the IP is more exhausted as compared to all the timepoints after infusion (as assessed from the negative Z scores). Moreover, both datasets are then in inconsistent with the findings in Fig. 4e, which suggests that cells are more exhausted after infusion, but the level of exhaustion does not decrease over time as suggested from the GSEA using Wherry 2007 gene set. Please explain this inconsistency in the discussion and possibly consider providing a list of gene sets in the supplementary.

Response: Thank you for the comments. The Wherry 2007 and Long 2015 exhaustion gene sets were established using different models of exhaustion – and therefore contain different genes. The Wherry 2007 gene set compares exhausted CD8⁺ T cells from chronic viral infection to functional CD8⁺ T cells after acute viral infection in mice. In contrast, the Long 2015 gene set compares exhausted CAR-T cells undergoing tonic signaling to functional CAR-T cells generated from healthy human donors. Using these two different gene sets, we acknowledge that the level of exhaustion in the infusion product compared to timepoints after infusion appears inconsistent. However, the only conclusion we can draw from these complementary gene sets is that an exhausted gene signature does not increase after infusion. Gene sets used in GSEA are listed in Supplementary Table 4 and we have modified the manuscript at on page 9, line 210-218 to better explain the differences between the two exhaustion gene sets used in the analysis as follows:

“We performed GSEA using two different exhaustion gene sets (Supplementary Table 4). One gene set compares exhausted CD8⁺ T cells from chronic LCMV infection to functional CD8⁺ T cells after acute infection in mice²⁴. The second gene set compares exhausted, tonically signaling GD2.28z CAR-T cells to functional GD2.BBz CAR-T cells generated from healthy human donors²⁵. Comparison with the first gene set showed a decrease in expression of the

exhaustion gene signature over time after infusion; whereas comparison with the second gene set did not show any changes in expression of the exhaustion gene signature over time (Fig. 4c). Overall, expression of an exhaustion gene signature did not consistently increase over time after infusion.”

Comment 5: In Fig. 5a, PCA analysis was provided only for cells that declined in blood over time after infusion. Please also consider doing the same for cells that progressively increased.

Response: Thank you for bringing this to our attention. We have revised the manuscript to improve clarity. Data from Figure 5a was not limited to cells that declined in blood over time after infusion. We performed PCA analysis of gene expression residuals on all CAR-T cells that were isolated from the IP and at the early, late, and very late time points. We have revised the manuscript on page 10, line 228-231 to better explain these data. We now state that:

“Principal component analysis (PCA) of gene expression residuals demonstrated marked heterogeneity in the transcriptional profiles of CD8⁺ CAR-T cells in the IP. After infusion, transcriptional heterogeneity of CD8⁺ CAR-T cells progressively declined in blood over time (Fig. 5a).”

Comment 6: On page 9, paragraph 2, please add references to Table 1 in the text.

Response: References to Table 1 have been added to page 11, line 258 and line 261 of the manuscript.

Response to Reviewer 2 Comments

Reviewer #2, clinical CAR T cell expert (Remarks to the Author):

Comment 1: Sheih et al analyzed the clonal selection in vivo of CD19.CAR-T cells after adoptive transfer and performed single-cell transcriptional profile of CAR-T cells. The authors focused on CD8⁺ T cells. The manuscript is an extension of previous observations published by the same group. Methodologically the manuscript is in general accurate, however the overall impact of the reported data is rather limited.

Response: Thank you for the comment. We agree that these data are an extension of – and consistent with – our previous data showing polyclonality of CAR-T cell expansion *in vivo*. In the current manuscript, we present a far more extensive analysis of clonotype kinetics using TCRB

gene sequencing and, in the revised manuscript, integration site analyses. To the best of our knowledge, these data have never been reported.

Major comments

Comment 2: The authors observed a contraction of the TCR beta repertoire of the CAR-T cells in samples collected at later time points as compared to early time points, at the peak of CAR-T cell expansion in vivo, which is largely expected.

Response: We agree that a contraction of the TCRB repertoire of the CAR-T cells might be expected given the decline in CAR-T cell numbers at later time points. However, we feel that loss of TCRB diversity (adjusted for the minimum number of recovered sequences at all timepoints) after infusion could not necessarily be predicted or confirmed without experimental data.

Comment 3: It remains unclear if the analysis of the TCR repertoire can help in identifying which are the characteristics of the CAR-T clones that are expanding/persisting better in vivo.

Response: We agree that analysis of the TCR repertoire in isolation cannot identify the characteristics of the CAR-T cell clones that are expanding/persisting better in vivo. Therefore, we performed single-cell RNA sequencing, which generated paired gene expression and TCR sequence data from single cells, to identify the transcriptional profile of CAR-T cell clones in the infusion product that relatively expanded after infusion (Figure 6). We showed that expanding clones mainly originated from two transcriptionally distinct clusters of CAR-T cells in the infusion product.

Comment 4: It may be possible, that in addition to specific functional characteristics of the T cells expressing the CAR (which are not identified in this manuscript), the clonal insertion of the vector may play a relevant role. This aspect should be experimentally analyzed.

Response: The reviewer raises an important point about the potential impact of the vector integration site on clonal expansion. A recent study reported a highly unusual case in which a single dominant CAR-T cell clone was driven by lentiviral integration into the TET2 gene¹. In light of this report and in response to comments from reviewers 2 and 3, we analyzed vector integration sites in CD8⁺ CAR-T cells isolated from the infusion product (IP) and blood from patients in our study (n=7) to determine if integration site is a key driver of clonal kinetics. Our findings are described in the revised results section (pages 6-8) as follows:

“We analyzed the lentiviral integration profile of CD8⁺ CAR-T cells isolated from the IP and from blood after adoptive transfer in ALL (n=3) and NHL (n=4) patients. A total of 55,382 unique integration events (sites) were identified across all patients and samples. The observed integration sites were consistent with lentiviral integration patterns previously described in human T cell lines²³. Approximately, 82.6% of all sites (45,771) were within genes and integration in introns was more frequent than in exons (**Supplementary Figure 2**). Consistent with our findings from TCRB sequencing, different clonotypes defined by integration site exhibited distinct *in vivo* kinetic patterns after CAR-T cell infusion (**Supplementary Figure 3**). We examined whether these changes in clonotype abundance were associated with distinct genomic loci of vector integration. We identified two different groups of integration sites, sites that either increased or decreased in relative abundance by at least 5-fold between the IP and after infusion. Comparison of genes harboring these integration sites revealed enrichment in many of the same biological pathways (**Supplementary Figure 4**). However, genes in biological pathways associated with lymphocyte activation, TCR signaling, and regulation of type 1 interferon were more enriched among genes harboring integration sites that increased in relative abundance; and genes associated with T cell differentiation and the cellular response to UV were more enriched among those harboring integration sites that decreased in relative abundance (**Supplementary Figure 4**). These data likely reflect differences in genes that are active at the time of lentiviral transduction.

A recent report identified dominance of a single infused CAR-T cell clone in a single patient associated with integration into the *TET2* gene. While integrations in the *TET2* gene were observed in our analyses (12 sites in 6 patients), none of these integration sites were among the top 20 most abundant sites identified in any patient or sample, indicating that integration within the *TET2* gene was not a key and frequent driver of clonal expansion in our study. Furthermore, in 2 patients with highly dominant TCRB clonotypes after infusion (ALL-2 and NHL-2), we did not identify single integration sites that were responsible for clonal dominance. No integration sites were found at a frequency as high as that of the dominant TCRB clonotype. The most dominant TCRB clonotypes in blood from ALL-2 and NHL-2 at the early time point were 46.0% and 16.8%, respectively. In contrast, in the same samples the highest frequency integration sites in each patient only represented 2.75% and 5.2% of the total integration sites, respectively. These data suggest that integration site is unlikely to be the key driver of clonal kinetics in our study.”

We have also modified the Method section on pages 17-19 and Discussion section on pages 13-14, lines 312-321.

Supplementary Figure 2. Lentiviral vector integration sites in CD8⁺ CAR-T cells. Bar chart depicts the percentage of integration sites found within an exon, intron, or not in a gene.

Supplementary Figure 3. Integration Site Analysis (ISA) suggests multiple clonal clones contribute to CAR-T cell pools *in vitro* and *in vivo*. Graphs represent the contribution (% frequency) of all identified clones by ISA in IP and blood samples collected after infusion (x-axis). Each color ribbon represents a unique clone demonstrating ≥ 1% frequency of sequence reads in a given sample. All other clones are grouped into the gray ribbon at the top of each graph. The total number of unique clones identified in the sample is listed underneath the sample ID for each graph (below the x-axis).

Supplementary Figure 4. Heatmap of biological processes identified using Gene Ontology (GO) enrichment analysis. Integration sites that either increased (Expanding) or decreased (Contracting) in relative abundance by at least 5-fold between the infusion product and in blood after adoptive transfer were identified and combined across all patients. The most significantly enriched GO categories in either expanding or contracting integration sites are plotted as a row of individual boxes and color coded based on FDR adjusted p-value.

References

1. Fraietta JA, Nobles CL, Sammons MA, et al. Disruption of TET2 promotes the therapeutic efficacy of CD19-targeted T cells. *Nature*. 2018;558(7709):307-312.

Comment 5: The performed analysis in samples of peripheral blood may not reflect what occurs in the bone marrow and lymph nodes. Are CAR-T cells in the peripheral blood, bone marrow and lymph nodes showing similar TCR repertoire at different time points?

Response: We acknowledge that although CAR-T cell counts in the blood correlate with elimination of disease in bone marrow and lymph nodes²⁻⁴, changes in the CAR-T cell repertoire in the peripheral blood may not reflect changes in the repertoire in the bone marrow and lymph nodes. Although we agree that studying CAR-T cells in the bone marrow and lymph nodes is important, we are not able to perform multiple serial marrow and lymph node biopsies from patients on these clinical trials to carry out these kinetic experiments. We have modified the manuscript on page 12, line 285-289 to discuss the need for additional studies on CAR-T cells at sites other than blood.

References

2. Turtle CJ, Hanafi L-A, Berger C, et al. Immunotherapy of non-Hodgkin's lymphoma with a defined ratio of CD8+ and CD4+ CD19-specific chimeric antigen receptor-modified T cells. *Sci Transl Med.* 2016;8(355):355ra116.
3. Turtle CJ, Hay KA, Hanafi L-A, et al. Durable Molecular Remissions in Chronic Lymphocytic Leukemia Treated With CD19-Specific Chimeric Antigen Receptor-Modified T Cells After Failure of Ibrutinib. *J Clin Oncol.* 2017;35(26):3010-3020.
4. Turtle CJ, Hanafi L, Berger C, et al. CD19 CAR-T cells of defined CD4+:CD8+ composition in adult B cell ALL patients. *J Clin Invest.* 2016;126(6):2123-2138.

Comment 6: The single-cell transcriptional profile shows that CAR-Ts are highly cytotoxic at early time points and became more “transcriptionally quiescent” at later time points during the contraction phase of CAR-T cells. This is largely expected and not particularly informative. It would be more relevant to assess the single-cell transcriptional profile of CAR-T cells at later time point in bone marrow, where these cells may remain more active since they continue eliminating normal B cells that the bone marrow is regenerating.

Response: The reviewer's point that CAR-T cells in the bone marrow might exhibit a different transcriptional profile is entirely valid; however, these samples are not available. We have modified the manuscript on page 13, line 298-299 to discuss the need for additional single-cell RNA sequencing studies on CAR-T cells at the site of tumor.

Comment 7: The single-cell transcriptional profile includes 2 patients with NHL and 2 patients with CLL. These are really different diseases in term of “T cell quality” as recently reported by

University of Pennsylvania. Furthermore, CAR-T cells for NHL and CLL were manufactured using two completely different approaches (TCM versus bulk T cells).

Response: To account for the differences in disease type and manufacturing, we performed DEG and GSEA analyses for each patient and only considered genes and gene sets that were differentially expressed between time points in all four patients. This allowed us to identify distinct transcriptional programs expressed by CAR-T cells at each time point, irrespective of disease type and manufacturing differences (Figure 4b).

Response to Reviewer 3 Comments

Reviewer #3, translational and clinical CAR T cell expert (Remarks to the Author):

Comment 1: Sheih and colleagues have conducted a comprehensive analysis of the persisting CD8 CD19 CAR T cells in patients with heme malignancies. The work is done using state of the art single cell profiling of the CAR T cells captured during manufacturing and at various times following infusion in patients.

Response: Thank you for the comment.

Comment 2: There are a number of inherent biases in this type of study, since it only looks at successful treatments, rather than in patients that have poor engraftment and no clinical responses.

Response: We understand the reviewer's point regarding the design of this study in which we selected patients with durable persistence of CAR-T cells. However, selection of patients with persisting CAR-T cells was necessary to study the clonal kinetics and transcriptional profiles of infused CAR-T cells. Using this approach, we were able to directly compare the transcriptional profiles of infused cells bearing clonotypes that either persisted or disappeared after adoptive transfer in individual patients. While we agree that it is of interest to compare the transcriptional profiles of CAR-T cells between patients with poor and good engraftment of CAR-T cells, this may not be feasible in the absence of tumor biopsies to enable adjustment for differences in their tumor microenvironment.

Comment 3: There is an implied bias/assumption that the persisting cells have preferred transcriptional states (i.e. cluster 2 and 4) and that the "loser" CARs have a cluster 1 signature.

It is possible that other factors might be involved such as cell extrinsic mechanisms such as immune responses arising to the CAR that cause rejection and a “loser” state.

Response: We agree that an immune response against the CAR might lead to rejection of CAR-T cells. However, given that all CAR-T cells in our study express the same immunogenic epitope within the scFv, we do not expect the immune response to differentially target distinct clonotypes. Furthermore, the transcriptional profiles did not show differences in endogenous antigen processing and presentation pathways that might render cells in clusters 1 and 3 more immunogenic. Additionally, the four patients analyzed in our scRNA-seq study exhibited robust CAR-T cell expansion and persistence, suggesting that an anti-CAR immune response is unlikely in the studied patients.

Comment 4: In addition, authors expand the CAR T during manufacturing with an EBV LCL line (?autologous), and don't state that host responses directed to CAR T with TCRs reactive to EBV antigens might be involved in persistence, independent of the transcriptional profiling. Authors should comment on this.

Response: We appreciate this insightful comment and agree with the reviewer that recognition of EBV antigens through the endogenous TCR might play a role in CAR-T cell persistence. However, the CD19⁺ EBV LCL cell line used during CAR-T manufacturing is not autologous. We have modified the manuscript on page 15, line 361 to clarify the manufacturing process. Thus, EBV-driven TCR-mediated expansion of CAR-T cells is unlikely given the absence of cross-presenting antigen presenting cells in the manufacturing process and absence of LCL cells in the infused products. Even in the uncommon setting of matching one or more HLA alleles between the LCL cell line and the patient, studies have reported that concurrent signaling through the endogenous TCR and CAR leads to increased apoptosis and clonal deletion of CD8⁺ CAR-T cells^{5,6}. These data indicate that the manufacturing process is unlikely to enrich for EBV-reactive CD8⁺ CAR-T cells.

References

5. Yang Y, Kohler ME, Chien CD, et al. TCR engagement negatively affects CD8 but not CD4 CAR T cell expansion and leukemic clearance. *Sci Transl Med.* 2017;9(417).
6. Ghosh A, Smith M, James SE, et al. Donor CD19 CAR T cells exert potent graft-versus-lymphoma activity with diminished graft-versus-host activity. *Nat Med.* 2017;23(2):242-249.

Comment 5: Figure 1d. Patient ALL-2 has a dominant clone at the late timepoint, also shown in Supple Fig 1. Where is the integrated CAR and is it near a gene that might be implicated in clonal T cell proliferation? Same question for patient NHL-2 clonotype who has emergence of clonal outgrowth.

Response: We agree with the reviewer that the integration site may impact clonal kinetics. To address this point, we performed integration site analyses on 7 patients (**see response to reviewer 2, comment 4**), including ALL-2 and NHL-2. With regards to ALL-2 and NHL-2, we have modified the Results on pages 7-8 as follows:

“...in 2 patients with highly dominant TCRB clonotypes after infusion (ALL-2 and NHL-2), we did not identify single integration sites that were responsible for clonal dominance. No integration sites were found at a frequency that was as high as that of the dominant TCRB clonotype. The most dominant TCRB clonotypes in blood from ALL-2 and NHL-2 at the early time point were 46.0% and 16.8%, respectively. In contrast, in the same samples the highest frequency integration sites in each patient only represented 2.75% and 5.2% of the total integration sites, respectively. These data suggest that integration site is unlikely to be the key driver of clonal kinetics in our study. “

Comment 6: Related to the above, authors previously reported that the peak CAR T in patients was polyclonal (Turtle, JCI 2016). This doesn't seem consistent with the oligoclonal outgrowths in patients ALL-2 and NHL-2.

Response: We agree that the oligoclonal CAR-T cell expansions in patients ALL-2 and NHL-2 differ from our previous report of polyclonality in CAR-T cells after adoptive transfer. However, the data highlights the variability in clonal kinetics that may be observed in different patients and the need to study the CAR-T cell repertoire to gain a better understanding of different patterns of *in vivo* behavior after adoptive transfer.

Comment 7: Figure 3c. why are only 6 patients shown in this panel? For example, why isn't data from patient ALL-2 shown in this panel?

Response: Thank you for pointing out this oversight. We have modified Figure 3c to include all patients except NHL-4. Patient NHL-4 was excluded from this figure because we do not have samples at a late time point from this patient. As a result, we cannot distinguish between clones that progressively increased and clones that transiently increased and declined in this patient.

Figure 3c

Comment 8: Page 15. “Following sequence alignment and filtering, data from cells with unique gene counts less than 200, a percentage of mitochondrial genes greater than 20% or greater than 40,000 UMIs were removed, resulting in 62,167 CD8⁺ single CAR-T cells for analysis.” Authors should state how many cells were discarded at this step so that the reader can judge how many cells were required to obtain 62,167 CD8 CAR T cells.

Response: A total of 2,340 cells were discarded at this step. We have added this information on page 20, line 486 of the manuscript.

Comment 9: Supple Fig 2. How do authors explain the decreased scFv CAR expression compared to CD3 and CD8A? At the protein level, there are approximately equal numbers of CAR and TCR/CD3 reads. Is silencing of the transgene occurring?

Response: Thank you for the question. We do not believe transgene silencing is occurring because CD8⁺ CAR-T cells were sorted to high purity based on protein expression of the transgene marker, EGFRt, which is separated from the scFv CAR by a T2A ribosomal skip sequence in a bicistronic vector. Expression of the scFv in the CAR appears to be decreased

compared to CD3 and CD8A because of the 5' single-cell RNA sequencing method (10x Genomics) used in this study. This method only sequences and aligns the 5' end of the mRNA transcript to the reference genome. As a result, CAR expression will not be detected if the transgene integrates in the middle or 3' end of the transcript, which leads to the appearance of lower CAR expression. We have modified the figure legend for Supplementary Figure 5 to explain why the 5' scRNA-seq method might not detect all scFv sequences of the CD19 CAR.

Response to Reviewer 4 Comments

Reviewer #4, expertise in T cell repertoire in immunotherapy (Remarks to the Author):

This study utilizes TCR beta chain sequencing to investigate the expansion and maintenance of anti-CD19 CAR T-cell clonotypes after transfer to patients with B-cell malignancies. Therefore, prevalence of TCR beta clonotypes is measured in the infusion product (IP) and at later time points in the patients receiving these products. In addition, CAR T-cell populations are subjected to single-cell RNA sequencing (scRNAseq) (4 populations sequenced) and further TCR beta sequencing (2 of the 4 populations) before and after infusion into patients. These datasets are then used to identify clonotypes that show increased (IRF) and decreased relative frequency (DRF) in patients vs. IPs. Finally, a set of differentially expressed genes in IRF clones vs. DRF clones is identified.

The main findings of the manuscript are 1) that clonotypes differ substantially in their expansion after infusion 2) that increased expansion after infusion correlates with differential gene expression already in the IP.

Comment 1: The technical effort made and the technical quality of the applied approaches is high—especially for a clinical setting. The basic observation, of changing clonotypic dominance is surprising and at least to my knowledge novel. Still, I feel that the manuscript, in its current form, lacks conceptual advance.

Response: We appreciate the reviewer's comment that our data is surprising and novel.

Major concerns:

To warrant publication in Nature Communications, I feel that two sets of questions would have to be addressed:

Comment 2: How do the distinct transcriptional clusters identified via scRNAseq in the IP align

developmentally? Is cluster 2 a precursor of cluster 1 or is it the other way around? Gaining more insight here, e.g. via pseudotime analyses, could lay the foundation for answering the second set of questions:

Response: We agree with the reviewer that understanding the developmental relationships between the four infusion product clusters is important. As suggested, we performed pseudotime analysis within each patient using Monocle but found that cells in clusters 1 and 2 overlapped along the pseudotime axis in all four patients. Using this approach, we were unable to determine any developmental relationships among the four clusters. This might be due to the fact that the four IP clusters were sampled from a single time point, whereas pseudotime analysis is typically used to study developmental datasets where there is a fine time sampling. Additionally, these clusters might be unrelated, and these distinct transcriptional phenotypes might be driven by stochastic effects related to T cell stimulation and activation during manufacturing. We have discussed these points on page 13 in line 304-307 of the manuscript.

Comment 3: Can the clusters/subsets found in the IP be preferentially induced by modulating the production process of CAR T cells? Do they preferentially arise at certain time points of the production process?

Response: The reviewer poses important questions regarding the development of IP clusters. The manufacturing process is complex and there are many factors that could be modified such as the starting T cell subset for transduction, the activation and cytokine conditions, and the expansion protocol. Given the large number of factors that would need to be tested and the amount of data that would need to be reported, we feel that this work is beyond the scope of this manuscript.

Minor concerns:

Comment 4: Further on, it may be helpful to limit the differential gene expression analysis to IRF and DRF clonotypes showing particularly distinct expansion behaviors (e.g. the three strongest expanding vs. the three strongest decreasing clones).

Response: Thank you for this suggestion. We limited the differential gene expression analysis to the top 25% IRF and DRF clones and found that the same genes were differentially expressed as when all IRF and DRF clones were considered.

Comment 5: On a technical note: Cluster 4, showing high Ki67, CDK signatures, should not be defined or treated as a subset but simply as single cells identified by chance in the S or G2M

phase. The authors are correct in stating that overrepresentation of certain clonotypes in this cluster is indicative of higher proliferation activity. But to efficiently perform pseudotime and detailed clustering analyses the authors may profit from correcting their dataset for the proliferation associated artefacts that generate cluster 4. Further on, Cluster 3, showing strong mitochondrial RNA signatures likely consists of (pre-)apoptotic cells. This cluster may as well disturb detailed cluster and pseudotime analyses.

Response: We appreciate this suggestion concerning clusters 3 and 4 and their possible effects on clustering and pseudotime analyses. However, in our study, CAR-T cells in clusters 3 and 4, representing pre-apoptotic and proliferating cells, are of biological interest. As part of the manufacturing process, CAR-T cells in the infusion product were activated and expanded. As a result, these clusters represent cells that are proliferating or dying in response to stimulation through the CAR, which could affect whether a specific CAR-T cell clone expands and contributes to the anti-tumor response after adoptive transfer. Indeed, we found that infused cells that expanded after infusion originated from CAR-T cells in cluster 4 representing cells with higher proliferation activity.

Reviewers' comments:

Reviewer #1 (Remarks to the Author):

All points have been satisfactorily addressed. I do not have any further revisions.

Reviewer #3 (Remarks to the Author):

No further comments, as revisions to the manuscript have addressed my concerns. Recommend acceptance!

Reviewer #4 (Remarks to the Author):

I thank Sheih et al. for their responses to my inquiries. Unfortunately, the authors do not address the concerns raised in my initial review sufficiently.

In my comment #2 (as labelled in the authors' response letter) I asked for deeper analyses of the developmental relationships of the identified clusters. More precisely, I asked the authors to perform pseudo-time analyses of their single-cell RNA sequencing (scRNAseq) dataset and stated in comment #5 that this type of analysis will require a correction of the dataset for cell-cycle artefacts (mainly cluster 4, Fig.5) and cautious treatment of cluster 3 (Fig. 5), which showed a pre-apoptotic gene signature. However, the authors performed pseudo-time analysis without taking these considerations into account and did not find a relevant trajectory. They did not include the associated analyses in their response letter to me and did not even mention the performed analyses in the revised version of their manuscript. The authors justify using an uncorrected dataset as follows: "However, in our study, CAR-T cells in clusters 3 and 4, representing pre-apoptotic and proliferating cells, are of biological interest." To make myself clear, I did not ask the authors to throw this information away. I only asked them to correct for cell-death, and cell-cycle artefacts to allow for adequate trajectory inference. With respect to communicating the nature of cluster 3 and 4 to the reader, I do not quite understand why the word apoptotic (in reference to cluster 3) does not appear once in the manuscript and cluster 4 is consistently described as a distinct cell subset, while it rather signifies a certain cell-cycle state (S/G2M) occupied by various cell subsets. Further on, in their response to comment #2 the authors state that pseudo-time analysis may not have yielded significant results because "the four IP clusters were sampled from a single time point, whereas pseudo-time analysis is typically used to study developmental datasets where there is a fine time sampling". This statement is incorrect. The whole purpose of pseudo-time analyses is to use single-cell resolved datasets gathered at one time-point to identify developmental trajectories within these datasets. This is why it's called pseudo-time: Individual cells are analysed at the same point in real time but are at situated different positions of a developmental process, and thereby at different positions in pseudo-time.

In my comment #3: I asked for wet lab experiments providing additional insight on how to more deliberately generate CAR-T-cell products showing optimal cluster composition. The authors consider such experiments to be outside the scope of their manuscript and do not attempt a single experiment to provide additional insight concerning this question. I, however, consider such experiments as essential for providing mechanistic insight in an otherwise descriptive manuscript.

Taken together, after revision, the manuscript remains technically sophisticated but still lacks conceptual advance. Thus, I would ask the authors to revisit my original comments #2, 3 and 5 and address these questions base on further analyses of their scRNAseq data and further experimentation.

Reviewer comment: I thank Sheih et al. for their responses to my inquiries. Unfortunately, the authors do not address the concerns raised in my initial review sufficiently.

In my comment #2 (as labelled in the authors' response letter) I asked for deeper analyses of the developmental relationships of the identified clusters. More precisely, I asked the authors to perform pseudo-time analyses of their single-cell RNA sequencing (scRNAseq) dataset and stated in comment #5 that this type of analysis will require a correction of the dataset for cell-cycle artefacts (mainly cluster 4, Fig.5) and cautious treatment of cluster 3 (Fig. 5), which showed a pre-apoptotic gene signature. However, the authors performed pseudo-time analysis without taking these considerations into account and did not find a relevant trajectory. They did not include the associated analyses in their response letter to me and did not even mention the performed analyses in the revised version of their manuscript. The authors justify using an uncorrected dataset as follows: "However, in our study, CAR-T cells in clusters 3 and 4, representing pre-apoptotic and proliferating cells, are of biological interest." To make myself clear, I did not ask the authors to throw this information away. I only asked them to correct for cell-death, and cell-cycle artefacts to allow for adequate trajectory inference.

Response: We apologize for not sufficiently addressing the reviewer's concerns in our previous response letter. The reviewer raises valid concerns regarding pseudotime analysis and helpfully suggested correcting the dataset for cell-cycle and cell-death artefacts prior to analysis. To address the reviewer's concerns, we have assigned a cell cycle score to each cell using the *CellCycleScoring* R function from the Seurat package. This function calculates S and G2M scores and predicts classification of each cell in either S, G2M, or the G1 phase. As the reviewer correctly stated, CAR-T cells in cluster 4 were comprised of single cells in the S or G2M phase (**Figure 1**). Using Monocle, we assigned the cell cycle scores, the number of detected molecules per cell (nUMI), and the percentage of mitochondrial genes as covariates in differential gene expression analysis between infusion product clusters¹. This corrected gene list was used to define a CAR-T cell's progress through the manufacturing process and as input for pseudotime analysis. We found a large degree of overlap between CAR-T cells from all four clusters without a clearly defined trajectory to indicate any developmental relationships among the four clusters (**Figure 2A**). A similar single-cell trajectory was found when we performed Monocle without correcting the dataset (**Figure 2B**).

Figure 1: The percentage of CD8⁺ CAR-T cells within each infusion product cluster that is predicted to be in either G1, S, or G2M phase.

Figure 2: Single-cell trajectory analyses were performed using Monocle (A) after correction of the dataset for cell cycle scores, nUMIs, and the percentage of mitochondrial genes or (B) without correction of the dataset. CD8⁺ CAR-T cells in the infusion product of all four patients were ordered in pseudotime and colored based on cluster assignment.

The beta version of Monocle 3 was recently released and re-engineered to analyze large single-cell datasets². To gain further insight into the developmental relationships among the four clusters, we also performed pseudotime analysis on the corrected dataset (cell cycle score, nUMI, and the percentage of mitochondrial genes as covariates; and patient used as alignment group) using the Monocle 3 R package. Figure 3a displays the four CAR-T cell clusters on a UMAP and the trajectories as black line segments. Using this method, we still were not able to identify a consistent relationship between the four clusters (**Figure 3A**). Similar results were obtained after performing the trajectory analysis without correcting the dataset (**Figure 3B**).

Finally, we performed RNA velocity analysis of CAR-T cells in the infusion product, in which RNA velocity was based on the ratio of unspliced to spliced mRNA transcripts. This analysis method did not allow for correction of the dataset for cell cycle and apoptotic gene signatures. This method has been reported to successfully predict future transcriptional states in a mouse embryo scRNA-seq dataset³. Figure 4 displays the RNA velocity of CAR-T cells in the infusion product as shown by the direction of the arrows. The RNA velocity field did not show strong directional flow from one cluster to the other (**Figure 4**). Additionally, RNA velocity was low in CAR-T cells (indicated by arrow length), which might suggest that these cells are undergoing more limited transcriptional changes. Although we cannot rule out the possibility of a relationship between CAR-T cell clusters in the infusion product, we were unable to identify these relationships using pseudotime and RNA velocity analysis methods.

Figure 3: CD8⁺ CAR-T cells in the infusion product of all four patients were analyzed using Monocle 3 (A) after correction of the dataset for cell cycle scores, nUMIs, and the percentage of mitochondrial genes or (B) without correction of the dataset. Individual cells are plotted on a UMAP and colored based on cluster assignment. The defined trajectories are shown as black line segments.

Figure 4: RNA velocity of CD8⁺ CAR-T cells in the infusion product were computed and visualized on t-SNE plot. Arrows indicate the direction of the future states.

We have modified the text of the manuscript on page 10 and line 237-243 to report our findings from pseudotime and RNA velocity analyses and added Figures 1, 3A, and 4 of this response as Supplementary Figures.

References

1. Trapnell C, Cacchiarelli D, Grimsby J, et al. The dynamics and regulators of cell fate decisions are revealed by pseudotemporal ordering of single cells. *Nat Biotechnol.* 2014;32(4):381-386.
2. Cao J, Spielmann M, Qiu X, et al. The single-cell transcriptional landscape of mammalian organogenesis. *Nature.* 2019;566(7745):496-502.

3. La Manno G, Soldatov R, Zeisel A, et al. RNA velocity of single cells. *Nature*. 2018;560(7719):494-498.

Reviewer comment: With respect to communicating the nature of cluster 3 and 4 to the reader, I do not quite understand why the word apoptotic (in reference to cluster 3) does not appear once in the manuscript and cluster 4 is consistently described as a distinct cell subset, while it rather signifies a certain cell-cycle state (S/G2M) occupied by various cell subsets.

Response: We thank the reviewer for the reminder that these clusters do not represent distinct cell subsets but rather represent CAR-T cells that are activated, apoptotic, or in the S/G2M phase of the cell cycle (**Figure 1**). We have modified the manuscript to no longer use the term “cell subset”. We have also modified the text on page 13 and line 313-317 to better describe the nature of these clusters to the reader as follows:

“Higher expression of mitochondrial genes in cluster 3 is indicative of pre-apoptotic cells, while higher expression of cell cycle-associated genes in cluster 4 signifies cells in the S/G2M phase of the cell cycle. These transcriptionally different cell states may in part be due to stochastic effects of T cell stimulation and activation during manufacturing, or different levels of CAR expression in individual cells.”

Reviewer comment: Further on, in their response to comment #2 the authors state that pseudo-time analysis may not have yielded significant results because “the four IP clusters were sampled from a single time point, whereas pseudo-time analysis is typically used to study developmental datasets where there is a fine time sampling”. This statement is incorrect. The whole purpose of pseudo-time analyses is to use single-cell resolved datasets gathered at one time-point to identify developmental trajectories within these datasets. This is why it’s called pseudo-time: Individual cells are analysed at the same point in real time but are at situated different positions of a developmental process, and thereby at different positions in pseudo-time.

Response: We apologize for the lack of clarity in our previous response.

Reviewer comment: In my comment #3: I asked for wet lab experiments providing additional insight on how to more deliberately generate CAR-T-cell products showing optimal cluster composition. The authors consider such experiments to be outside the scope of their manuscript and do not attempt a single experiment to provide additional insight concerning this question. I, however, consider such experiments as essential for providing mechanistic insight in an otherwise descriptive manuscript.

Response: We agree that studies addressing manufacturing strategies to produce optimized CAR-T cell products are important but agree with the Editor that this is of lower priority for this manuscript. We have modified the manuscript on page 13, line 317-319 to highlight the need for these studies.

Reviewer comment: Taken together, after revision, the manuscript remains technically sophisticated but still lacks conceptual advance. Thus, I would ask the authors to revisit my original comments #2, 3 and 5 and address these questions base on further analyses of their scRNAseq data and further experimentation.

Response: We appreciate the reviewer’s comments and feel the manuscript is much improved as a result.

REVIEWERS' COMMENTS:

Reviewer #4 (Remarks to the Author):

I thank the authors for the re-review of their manuscript. I can follow the authors' argument that additional experiments investigating changes in the manufacturing procedure are outside the scope of this manuscript. All my other concerns have been sufficiently addressed. I recommend the manuscript by Sheih et al. for publication in Nature Communications.